# Nonlinear Optical Characterization of 2D Materials

**DOI:** 10.3390/nano10112263

**Published:** 2020-11-16

**Authors:** Linlin Zhou, Huange Fu, Ting Lv, Chengbo Wang, Hui Gao, Daqian Li, Leimin Deng, Wei Xiong

**Affiliations:** Wuhan National Laboratory for Optoelectronics, School of Optical and Electronic Information, Huazhong University of Science and Technology, Wuhan 430074, China; zll111218@163.com (L.Z.); fu_huange@163.com (H.F.); 18843174953@163.com (T.L.); m201972890@hust.edu.cn (C.W.); snail.hui@163.com (H.G.); m202073074@hust.edu.cn (D.L.); dlm@hust.edu.cn (L.D.)

**Keywords:** two-dimensional materials, nonlinear optics, nonlinear optical characterization, multi-modal spectroscopy

## Abstract

Characterizing the physical and chemical properties of two-dimensional (2D) materials is of great significance for performance analysis and functional device applications. As a powerful characterization method, nonlinear optics (NLO) spectroscopy has been widely used in the characterization of 2D materials. Here, we summarize the research progress of NLO in 2D materials characterization. First, we introduce the principles of NLO and common detection methods. Second, we introduce the recent research progress on the NLO characterization of several important properties of 2D materials, including the number of layers, crystal orientation, crystal phase, defects, chemical specificity, strain, chemical dynamics, and ultrafast dynamics of excitons and phonons, aiming to provide a comprehensive review on laser-based characterization for exploring 2D material properties. Finally, the future development trends, challenges of advanced equipment construction, and issues of signal modulation are discussed. In particular, we also discuss the machine learning and stimulated Raman scattering (SRS) technologies which are expected to provide promising opportunities for 2D material characterization.

## 1. Introduction

In 2004, Konstantin Novoselov and Andre Geim used Scotch tape to strip graphene with monoatomic thickness from graphite [1], marking the creation of two-dimensional (2D) materials for the first time. Graphene has attracted much attention because of its unique physical and chemical properties. In a short period of time, the 2D materials family has continued growing and developing. In addition to graphene, there are also representative transition metal chalcogenides (MX_2_, M = molybdenum, tungsten; X = S, Se), main group metal chalcogenides (GaS, InSe, SnS, SnS_2_, etc.) and many other 2D materials, covering materials types such as superconductors, metals, semi-metals, semiconductors, insulators. These 2D materials have different band structures and exhibit unique optical, electrical, and mechanical properties, providing a material basis for subversive innovation in many fields such as optoelectronics, catalysis, energy storage conversion, sensors, and biomedicine [2,3,4,5]. The accurate characterization of various physical and chemical properties of 2D materials is crucial in the functional device applications, such as layer number, crystal structure, crystal phase, defects, strain, chemical specificity. Commonly used characterization techniques include Raman microscopy, scanning electron microscopy (SEM), transmission electron microscopy (TEM), X-ray diffraction (XRD), atomic force microscopy (AFM), and fluorescence microscopy. Although the as-mentioned diverse characterization methods provide us with powerful research tools, the problem is that each method has its own advantages and disadvantages, and it requires multiple methods and cumbersome processes to achieve the comprehensive examination of the properties of 2D materials.

As a branch of modern optics, nonlinear optics (NLO) has developed into a key technology, which has been widely used in material characterization, light generation, quantum optics, and other fields [6]. It is very convenient to combine NLO and microscope techniques to image 2D materials. Generally, strong nonlinear signals can be acquired through the excitation of tens of milliwatts of laser energy, which enables multi-modal fast scanning to be completed in a short exposure time. High spatial resolution can be achieved by combining with a high numerical aperture objective lens further. The current research on the NLO response of 2D materials can be divided into two aspects. On one hand, it aims to apply a nonlinear response for the fabrication of various functional devices. On the other hand, the NLO response is used as a means of 2D material characterization. With the development of research in recent years, NLO has shown its potential as a powerful characterization tool. For example, second-harmonic generation (SHG) is one of the well-known NLO characterization techniques, which is widely used in the research of crystal axis orientation and other related directions. The application of third-harmonic generation (THG) and four-wave mixing (FWM) techniques in determining the layer numbers and anisotropy of crystallinity are also explored. Furthermore, some new nonlinear techniques, such as coherent Raman spectroscopy, have recently attracted increasing attention due to their superior sensitivity to the chemical specificity of materials, which gradually becomes a promising candidate to replace the spontaneous Raman spectroscopy.

At present, the characterization of various properties of 2D materials is normally implemented in collaboration with multiple techniques, because there is no perfect characterization method to unveil the material properties in every aspect [7]. Meanwhile, NLO provides us with an idea to simplify the problem for 2D material characterization. Therefore, it is necessary and important to review the recent progress of the NLO characterization of 2D materials and discuss how to choose a suitable NLO technique rationally for specific material property characterization.

In this paper, we will focus on how NLO achieve the rapid and accurate characterization of various properties of 2D materials. In the second part, we will briefly introduce the principles of nonlinear optics, common detection techniques, and typical NLO characterization systems. In the third part, we will focus on a series of important properties of 2D materials, which mainly includes the number of layers, crystal orientation, crystal phase, defects, strain and chemical reaction monitoring, material chemical specificity, and ultrafast dynamics of excitons and phonons. The detailed NLO characterization techniques of these properties are discussed one by one, which provides ideas for the in-depth study of 2D materials. In the fourth part, we discuss the potential of the NLO characterization technologies, the challenges and opportunities in constructing NLO characterization equipment, and the internal and external modulation of NLO signals. In addition, the applications of machine learning and stimulated Raman scattering (SRS) technology in 2D material characterization are also prospected.

## 2. Principle of NLO Microscopy Techniques

### 2.1. Nonlinear Polarization

From a macro perspective, the general relationship between the polarization P of the interaction between light and materials and the electric field strength E of the incident light can often be described as follows [8,9]:(1)$$P=\varepsilon_{0}\left(\chi^{\left(1\right)}E+\chi^{\left(2\right)}E^{2}+\chi^{\left(3\right)}E^{3}+\cdots \right)$$
where $\varepsilon_{0}$ is the vacuum permittivity and $\chi^{\left(n\right)}$ is the n-th-order NLO susceptibility. The first term describes the conventional (i.e., linear) optical effects, such as refraction and absorption, while the remaining terms are nonlinear parts. As a result of the small high-order nonlinear susceptibility, P and E are approximately linear when E is small. However, when the electric field is strong enough, the high-order (n≥2) terms become significant. Therefore, Franken et al. [10] did not observe the generation of second harmonics for the first time until the laser was invented in 1961. From a microscopic perspective, optical nonlinearity can only be expressed when the electric field E of the incident light is equivalent to the interatomic electric field with a typical value of 10^5^–10^8^ V/m. Therefore, an intense excitation source is required to generate an NLO effect.

The second-order nonlinearity (three wave mixing) is determined by $\chi^{\left(2\right)}$ of the second term in Equation (1), including SHG, sum and difference frequency generation (SFG, DFG), and optical parametric interaction (optical parametric amplification, OPA/optical parametric oscillation, OPO), which is used to amplify and generate a wide frequency range of pulsed laser, optical rectification effect, and Pockels effect.

Third-order nonlinear effects include THG, FWM, intensity-dependent refractive index change (optical Kerr effect and saturable absorption (SA)), and two-photon excitation fluorescence (TPEF), which usually arise from the third-order NLO susceptibility $\chi^{\left(3\right)}$.

The high-order NLO susceptibility $\chi^{\left(n\right)}$ is representative of the high-order multiphoton scattering/absorption/luminescence and high harmonic generation (HHG). Since the interaction intensity of nonlinear processes usually decreases with n [8,9], the high-order NLO effects (including HHG) are very low in intensity and not significant. Meanwhile, there are few applications for them in the characterization of 2D materials, so it will not be described in detail in this paper due to the length limitation. We will focus on the second- and third-order NLO effects, which are dominant and the most commonly observed phenomena in the nonlinear processes.

The satisfaction of the phase-matching condition is very important for the generation of NLO effects. If the phase is mismatched, the intensity of the nonlinear optical signal would be weak or even disappear. According to quantum mechanics, when the momentum and energy of the photon are conserved simultaneously before and after the nonlinear process, the phase-matching condition is satisfied. This condition is very strict for bulk crystals [9]. To optimize the nonlinear effect, the path of the incident light and crystal orientation must be carefully designed. However, if the size of the medium is reduced to the sub-wavelength range and is shorter than the coherence length, the phase-matching condition is easily reached for achieving strong nonlinear effects, which has been demonstrated in nanoscale metamaterials [11,12].

In addition, nonlinear processes are also sensitive to the crystal structure of the material. Due to the high tensor levels of $\chi^{\left(2\right)}$ and $\chi^{\left(3\right)}$, even if the linear optical response is isotropic, the NLO response of the material can also be anisotropic [13], which enables the application of the NLO effect to characterize the crystal orientation with ultra-high sensitivity [8,9]. If the medium is centrosymmetric, P(−E)=−P(E) is required. Then, the polarization P must be an odd function of  E. Therefore, according to Formula (1), all even-order NLO susceptibility will disappear in the centrosymmetric medium. Ideally, a second-order nonlinear signal, such as SHG, cannot be observed in media with inversion symmetry (such as liquids, gases, amorphous solids, and many crystals). Nevertheless, due to the discontinuity, the inversion symmetry may be broken on the surfaces or edges and produce a weak SHG signal. In contrast, odd-order nonlinearity (third-order nonlinearity) is allowed in any material, regardless of whether the material is centrosymmetric, such as THG and FWM [9].

Nowadays, nonlinear optics has become a very useful technology, which is applied to the imaging characterization of materials (including 2D materials), valleytronic applications, frequency conversion to generate new light sources, such as ultraviolet light sources, and it can also be applied to ultrashort pulse generation, optical parameter generation (OPG), THz generation, quantum optics, strong field and attosecond physics, and sensing fields, as shown in Figure 1 [14].

### 2.2. Nonlinear Optical Characterization Methods

Traditional 2D material characterization methods include optical light interferometry and ellipsometry, XRD, SEM, and AFM. However, compared to traditional characterization methods that are complex and time-consuming, NLO characterization methods have many obvious advantages of non-invasive detection, large characterization area, high-speed mapping, and quantitative characterization. Therefore, NLO characterization is a suitable and powerful tool for the characterization of 2D layered materials (2DLM) [7,15,16,17].

#### 2.2.1. Second-Harmonic Generation (SHG)

It is known that sum-frequency generation (SFG) is a second-order nonlinear process in which two incident photons are converted into another emitted photon, and the energy of the photon is exactly equal to the sum of the two excited photons. The SHG is the degenerate process of the SFG when $\;\omega_{1}=\omega_{2}$ [9]. SHG also belongs to the second-order nonlinear effect, which was first discovered in crystalline quartz by Franken et al. [10] in 1961. When the incident wave or photon has the same frequency, two photons with frequency ω interact in the medium and are converted into photons with frequency 2ω, which results in the generation of the second harmonics. The energy diagram is shown in Figure 2a, which satisfies the formula $\;\omega_{SHG}=2\omega$ [8,9]. This process is mediated by the transition through a virtual energy state. The virtual state is an intermediate quantum state and cannot be physically occupied, but many optical processes that are not originally allowed can be realized by virtual states. SHG is one of the most studied techniques in material characterization. Since SHG is a second-order nonlinear process, it cannot be generated in a medium whose internal structure exhibits centered inversion symmetry. However, due to surface effects (i.e., the lack of symmetry on the surface of the material), a weak SHG signal can also be observed in inversion symmetric materials, which is called surface SHG.

#### 2.2.2. Third-Harmonic Generation (THG)

THG is one of the kinds of third-order nonlinear effect. The generation process of the third harmonic is similar to SHG, but it is the interaction of three photons with frequency ω inside the medium and converted into one photon with frequency 3ω, that is, $\omega_{THG}=3\omega$. Figure 2b shows the energy diagram of THG. THG does not require centered inversion symmetry breaking, but it depends on the third-order NLO susceptibility of a medium and finite phase-matching in the excitation region. Therefore, no matter whether the material is centrosymmetric or not, THG may be generated. Its function is similar to SHG, while THG has a cubic dependence on the excitation light intensity [8,9].

#### 2.2.3. Four-Wave Mixing (FWM)

FWM is a third-order nonlinearity in which two or three photons of different wavelengths interact to produce two or one photon with new wavelengths. FWM is a third-order NLO process, which is independent of whether the material is inversion symmetric. Therefore, in theory, FWM can be generated in any material. Generally, FWM contains many different forms, one of which is shown in Figure 2c. This energy diagram satisfies formula $\omega_{4}=\omega_{1}+\omega_{2}-\omega_{3}$. FWM is widely used in various applications, such as imaging, wavelength conversion, and many other optical processing applications [8,9].

#### 2.2.4. Coherent Raman Spectroscopy

In spontaneous Raman scattering, a monochromatic laser beam of frequency $\omega_{p}$ enters the sample. Due to inelastic scattering, Stokes and anti-Stokes signals with frequencies $\omega_{s}$ and $\omega_{as}$ are generated, respectively [9]. Raman scattering is also an NLO process, and it is a very powerful tool in identifying the crystal structure, lattice vibration, strain, and other characteristics of 2D materials. Since there are already some excellent review papers discussing the spontaneous Raman spectroscopy [18,19,20,21], we will focus on the coherent Raman spectroscopy techniques in this review.

1.Coherent Anti-Stokes Raman Scattering (CARS)

Coherent anti-Stokes Raman scattering (CARS) belongs to a special case of FWM, which is a third-order nonlinear phenomenon. The main difference between CARS and FWM is that the resonance enhancement of CARS is generated by the vibrational transition. CARS is mainly used in physics, chemistry, and other related fields. The energy diagram is shown in Figure 2d. This process is an example of an optical parametric process. Pump beam $\omega_{p}$ and Stokes beam $\omega_{s}$ with different frequencies are usually used to excite the sample. When the frequency difference of the pump light and Stokes light is equal to the vibration frequency of the intrinsic vibration mode of the molecule, it can produce an anti-Stokes signal $\omega_{as}$ under the action of another probe light $\omega_{p^{\prime }}$, which satisfies the formula $\omega_{as}=\omega_{p}-\omega_{s}+\omega_{p^{\prime }}$. Usually, the probe beam and the pump beam are fixed at the same frequency, i.e., $\omega_{p^{\prime }}=\omega_{p}$. Then, the above formula becomes $\omega_{as}=2\omega_{p}-\omega_{s}$ [22]. CARS was discovered by Maker and Terhune in 1965 [23]; however, it was not applied to spectroscopy until 1974 because of the development of high peak power tunable lasers [24]. As a coherent process, the scattering cross-section of CARS is several orders of magnitude larger than the spontaneous Raman scattering, which is very suitable for high-speed imaging [25,26]. The image resolution of CARS is significantly better than traditional molecular Raman microscope. In addition, the distribution of molecular bonds can be measured very clearly. CARS has the characteristics of chemical selectivity, high sensitivity, high spatial resolution, and fast data acquisition rate, and it shows a quadratic dependence and a linear dependence of the intensities of the pump light and Stokes light, respectively.

2.Stimulated Raman Scattering (SRS)

SRS is a process that is caused by two photons, which excite the atomic vibration of energy level, and it is also a special case of FWM with a third-order nonlinear effect. The energy diagram is shown in Figure 2e. SRS was first observed in 1962 [27], but it has only been applied to microscopy in the last 10 years [28,29]. When the frequency difference $\Delta \omega =\omega_{p}-\omega_{s}$ (also known as Raman shift) matches a specific molecular vibration frequency, the Raman scattering process is significantly enhanced when the pump and Stokes photons are applied at the same time, and a certain amount of pump photons are converted into Stokes photons. The loss of the pump photons is called the stimulated Raman loss (SRL), and the gain of the Stokes photons is called the stimulated Raman gain (SRG). On the contrary, when Δω does not match the vibration frequency, SRL and SRG will not appear. Both SRG and SRL are linearly dependent on the intensities of the pump light and Stokes light. This process is an energy transfer process and does not generate photons with new wavelengths. SRS microscopy has become a better alternative to CARS microscopy due to the elimination of spectral distortion and nonresonant background, simple and clear contrast interpretation, and linear intensity dependence on the analyte concentration, which enables an accurate and quantitative analysis of 2D materials directly [28,29,30].

#### 2.2.5. Two-Photon Excitation Fluorescence (TPEF)

TPEF microscopy was first demonstrated by Denk et al. [31] in 1990. If the energy of two photons is close to a certain transition frequency of the target molecule, the target molecule will absorb the two photons to reach the excited electronic state and then emit a single fluorescent photon with a higher energy than any incident photon, which is called TPEF [31,32]. The energy diagram is shown in Figure 2f. TPEF microscopy is an important fluorescence imaging technique. Unlike traditional fluorescence microscopy (the emission wavelength is longer than the excitation wavelength), the wavelength of its emitted light is shorter than that of two excitation photons. At the same time, the excitation volume of TPEF is small, which greatly reduces the overall photo damage of the sample and reduces the photobleaching of the fluorophore. The longer incident wavelength in TPEF can improve the deep penetration ability in the tissue to about 1 mm, which can reduce photolysis damage. Due to the secondary dependence of the TPEF signal on the intensity of the excitation light, it essentially provides the capability of optical slicing. Due to multiphoton absorption, the background is strongly suppressed. The TPEF wavelength will be greater than the SHG wavelength. Experiments have shown that TPEF and SHG images have almost the same signal quality [33].

#### 2.2.6. Typical Instrument

1.Typical SHG/THG Instrument

As shown in Figure 3a, the continuous wave laser is not strong enough to excite the materials with a large electric field between atoms, so an ultrashort pulsed laser can be used as the excitation source. The ultrashort pulsed laser can provide very high peak power, so it is used to generate NLO signals, and the average power is low enough to avoid damage to the sample.

The half-wave plate is used to control the polarization of the incident light, and polarization-dependent signal (such as SHG, THG) detection can be performed. An attenuator can be added at the front to control the laser incident intensity. The high NA objective lens is used to focus the beam to obtain a higher intensity incident field. The signal intensity of the nonlinear process is exponentially proportional to the incident field intensity, and the incident field intensity attenuates with the square of the axial distance from the focal point. This effect makes the nonlinear optical microscope have the same intrinsic three-dimensional slicing ability as the confocal microscope [34,35].

The galvanometer is used for beam scanning. Raster scanning is performed by the galvanometer, and the excitation beam is scanned on the sample. After the above process, an image is obtained by collecting an emission signal as a function of position. A three-dimensional stage can be also used for scanning detection as an alternate of a galvanometer. There is no difference in the optical properties of the system [32]. However, the beam galvanometer scanning method is generally considered to be faster in terms of the imaging speed [29]. Finally, a filter is used to separate the generated NLO signal from the excitation beam.

There are many types of photodetectors, such as photomultiplier (PMT), avalanche photon diode (APD), charge-coupled device (CCD), or spectrometer. Different photodetectors can be selected according to actual needs. PMT with high sensitivity can be selected for detection when the signal is weak. At the same time, both PMT and APD have a fast response speed, which are suitable for situations that require fast detection. CCD is a good choice when a graph of signal intensity needs to be acquired. A spectrometer is needed if further analysis needs to be performed by detecting the spectrum of the signal, for example, to determine the composition of the material by the changes in the wavelength and intensity of the signal.

2.Typical CARS/SRS Instrument

As shown in Figure 3b, CARS/SRS microscopies have strict requirements for excitation sources. First, an ultrashort pulsed laser is required to provide high excitation fields, and study has shown that the pulse width is about a few picoseconds [22]. Second, a dual-wavelength laser is required to provide pump light and Stokes light, and its frequency difference should be tunable to cover a series of vibration modes found in the sample. Third, it is better to choose near-infrared excitation to minimize the sample damage and maximize the depth of penetration [36].

Effective CARS/SRS generation requires perfect time synchronization between the Stokes and the pump pulse sequence. Therefore, an optical delay line is required to precisely adjust the time synchronization of the two beams of pulsed laser [37,38].

Compared with CARS, the SRS microscopy does not detect the signal at a new wavelength, but it does detect the intensity change of the excitation light. The SRS process needs to modulate the Stokes beam, so an optical modulator is required in front of the Stokes optical path. At the same time, the subsequent signal detection also needs to use a lock-in amplifier to demodulate and extract the SRS signal, as shown in the yellow dashed box in Figure 3. If the signal is weak, a preamplifier is also needed in addition to the lock-in amplifier to amplify the weak signal [28,29].

In terms of detecting signals, a forward-detected CARS (F-CARS) microscope is suitable for imaging objects with a size equal to or greater than the excitation wavelength. For smaller objects, F-CARS is not suitable, because the contrast is limited by the larger nonresonant background in the solvent. The backward-detected CARS (epi-detected CARS, E-CARS) microscope provides a sensitive method for imaging objects whose axial length is much shorter than the excitation wavelength, because it can eliminate large background noise from the solvent. The condenser lens (or objective lens) is used to collect the forward CARS signal. The forward signal is the superposition of CARS radiation from the scatterer and CARS radiation from the solvent, and it is always highly directional. The backward CARS signal is collected with the same objective, which is set to focus the laser beams. A dichroic beam splitter is used to separate excitation light and signal light [22].

SRS generally uses the forward detection method, because it detects the intensity loss of the transmitted pump light, which propagates in the forward direction. The reason why CARS can easily detect the backward signal is that the method essentially generates a backward signal of the new frequency. However, since the scattering of many samples is greater than the absorption, and the beam has been scattered multiple times, a large part of the forward pump and Stokes beam are redirected to the backward direction; thus, the backward detection can also be performed. Through a polarizing beam-splitter and an achromatic quarter-wave plate, the excited backward signal can be separated. However, the backward detection method is only suitable for high-scattering samples, and the intensity is still very low (about 1%) compared to the forward detection, so the image collected during the same time has a lower signal-to-noise ratio and it is usually difficult to achieve real time imaging [26,28].

## 3. Characterization of 2D Material Properties

### 3.1. Number of Layers

2D materials are special kinds of crystals with atomic thickness, which have unique electrical, optical, mechanical, chemical, and thermal properties and can often be used in the construction of high-performance devices. Determining the layer number of 2D materials is crucial for device fabrication. In traditional characterization methods, AFM, SEM, TEM, and scanning tunneling microscope (STM) are often used to determine the number of layers of 2D materials. However, in addition to being time-consuming, these methods generally require a costly operating environment and material substrate, and they may cause damage to the sample itself. In recent years, with the rapid development of NLO of 2D materials, the dependence of the material’s nonlinear optical effects on its layers has gradually been discovered. According to the current research, the number of layers of 2D materials is usually characterized by SHG and FWM. There is also a small amount of studies on the relationship between THG and layer numbers. Combining a variety of nonlinear optical characterization methods, we can also perform multi-modal characterization on samples to determine the number of layers more accurately.

In general, only 2D materials with broken inversion symmetry have SHG signals. Although in actual research, we can still observe little SHG in the case of even layers (this is due to the surface effect of the material). In case that the material structure is not destroyed, when the number of layers of the 2D materials is odd, the SHG intensity of some materials (such as WSe_2_ and WS_2_ [39], MoS_2_ [17]) decreases with the increase of the number of layers, which is resulted from the thicker sample reabsorbing more SHG photons when the energy of the SHG photon is greater than the band gap [14], as shown in Figure 4a,b. The relationship between the SHG intensity and the number of layers is nonlinear, because in general, if the thickness of sample layers is much smaller than the coherence length, the SHG intensity would have a quadratic dependence on the film layers [40]. For other materials (such as ReS_2_ [41], GaSe [42], InSe [43]), their SHG strength increases with the increase of the number of layers as shown in Figure 4c,d. For the layered metal mono-chalcogenide III–VI semiconductor, its nonlinear coefficient changes when the material is thin (within five layers), which makes the material layers and SHG strength have a cubic relationship by power-law fitting. Some materials such as MoTe_2_ [44], as the number of layers increases, the SHG intensity increases first and then decreases, as shown in Figure 4e. When the number of layers is less than five, the band gap of MoTe_2_ will change from direct to indirect. So, its light absorption decreases with the increase of the number of layers, which leads to an increase in the intensity of SHG. When the number of layers is more than 5, MoTe_2_ becomes an indirect band gap semiconductor with weaker light absorption, but the total attenuation of its SHG intensity is also increased because of the increased number of layers. Some 2D materials also output a strong SHG signal in the case of even layers for various reasons such as stacking or antiferromagnetic behavior (e.g., layered antiferromagnetic material CrI_3_) [45].

It is known that the generation of an FWM signal does not depend on the broken inversion symmetry of the material, so it exists in almost all materials. Therefore, we can also determine the number of layers by the FWM signal. Figure 5a shows the linear relationship between the FWM signal of MoS_2_ and its number of layers [17]. Although the SHG system is simple in operation and has a high signal intensity, it cannot detect even number–layer materials. So, FWM is applied to explore the relationship between the layers of a 2D material and the nonlinear signal strength. Figure 5b shows the linear relationship between a few layers of (N ≤ 6) graphene flake and the FWM signal strength [46,47] (some studies have shown that when the number of graphene layers is N ≤ 13, the relationship between FWM strength and the number of layers is always linear [14]). As a very useful characterization technique, we believe that FWM has good development prospects. At present, the application of CRS characterization in the small Raman wavenumber region is still limited. Only graphene and hexagonal boron nitride (h-BN) with large Raman wave numbers are reported to have the CRS characterization results by far. Figure 5c shows the dependence of CRS on the number of h-BN layers [48].

Another effective technique of measuring the number of layers is THG. The above materials using SHG and FWM to measure the number of layers, such as MoS_2_ (the number of layers of which has a quadratic dependence on the THG intensity) [49] and graphene [50], also have a THG signal, as shown in Figure 5d,e. However, compared with SHG and FWM, THG is not commonly used, because the THG strength of most 2D materials is much smaller than that of SHG. Meanwhile, black phosphorus (BP) is also a well-known 2D material. Unlike other 2D materials, the THG response radiated from BP is very sensitive and highly dependent on the polarization of excitation due to the anisotropy and the interference effect between the surface and volume nonlinear contributions. Since BP is centrosymmetric, SHG cannot be used for the characterization under normal circumstances. As the conversion efficiency of FWM is much lower than THG [51,52], the contrast of THG is higher than FWM signal; therefore, its layer number is usually characterized by THG [53,54,55], as shown in Figure 5f. The non-monotonic relationship between the THG intensity and the number of layers is caused by the interference between the BP surface and the nonlinear contribution of the substrate.

In summary, there are several NLO techniques to be used for characterizing the number of material layers. SHG can be used to judge the parity layer due to its dependence on the broken central inversion symmetry. FWM is used to study the dependence between the material layers and the nonlinear signal strength, which usually has a good linear relationship when the number of layers is relatively small, but the relationship between the FWM signal and the number of material layers is not always linear. The THG signal is chosen to characterize the number of the BP layers because the conversion efficiency of the THG signal is much higher than that of the FWM. If you intend to use nonlinear optics to characterize the layer number of 2D materials, you need to first determine which NLO characterization technique can be used according to the crystal structure of the material. In view of the advantages of each nonlinear optical characterization technique, the multi-modal NLO characterization of 2D materials has become a hot topic. For example, we can combine SHG and FWM to characterize the layer number of 2D materials by judging and analyzing the parity of the number of layers and the strength of the nonlinear signal [17].

### 3.2. Crystal Structure Symmetry

#### 3.2.1. Crystal Orientation

The crystal structure has a three-dimensional periodicity in spatial arrangement, and each crystal variety can provide itself with a set of natural and reasonable crystal axis systems containing three crystal axes. Determining the crystal orientation is crucial for studying the growth and application of 2D materials. Common characterization methods include angle-resolved Raman spectroscopy, X-ray diffraction, high-resolution transmission electron microscopy (HRTEM), and STM, but these techniques not only operate in a complicated manner, they may also cause certain damage to the sample. In addition, Raman scattering has limited applicability to manifest the crystal orientations. For examples of graphene, MoS_2_, and other isotropic 2D materials, the intensity of the Raman characteristic peak does not change with the angle between the polarization direction of the incident light and the lattice direction. With the development of NLO inspection, the advantages of high speed and non-destructive characterization have made it a powerful method for determining the orientation of the crystal axis.

For 2D materials with inversion symmetry broken, such as MoS_2_ and WS_2_ [56], it is found that the SHG signal can only be observed in odd-numbered layers, but it disappears in even-numbered layers. In the early research, the strong SHG response of the odd layer was used to effectively determine the crystal axis orientation of the crystal through the polarized SHG method. It was found that the SHG signal of the 2H phase MoS_2_ is 6-fold symmetric (Figure 6a) [57]. The direction of the minimum intensity of SHG is parallel to the zigzag of Mo-Mo. Using this method, the orientation of the crystal axis of MoS_2_ grown in a large area can be quickly judged (Figure 6b) [58,59]. Similarly, the SHG response of WSe_2_ has a similar symmetry [60].

As the most advanced 2D transition metal sulfides, MoTe_2_, WTe_2_, ReS_2_, and ReSe_2_ show amazing potential and device performance in fields such as field effect transistors, photodetectors, piezoelectric materials, and thermoelectric materials. Due to the inversion symmetry of its single layer, the SHG response exhibits completely different characteristics from MoS_2_ and WS_2_. The even-numbered layers belong to the non-central symmetric C_1_ space group and show strong SHG signals [41,44,61].

The energy band structure and performance of manually stacked homogeneous and heterogeneous structures mainly depend on their symmetry. The SHG technique can be used to quickly characterize the orientation of the crystal axis of the heterojunction and to determine the twist angle of the two-layer material. Wei-Ting Hsu et al. [62] determined that the SHG from the twisted bilayer is a coherent superposition of the SH fields from each layer, and its phase difference depends on the stacking angle (Figure 6c). Similarly, Xin-Quan Zhang et al. [63] also used SHG technology to characterize the orientation of the crystal axis of the heterojunction to determine the twist angle of the two-layer material. The intensity of the parallel polarization component of SHG can be described as $I=I_{0}\cos^{2}\left(3\theta \right)$, where $I_{0}\;$ is the maximum SHG intensity, and *θ* is the angle between the polarization state of the incident beam and the direction of the armchair (Figure 6d). Unlike the usual 2H stacked structure whose SHG signal of an even number of layers disappears, Fan et al. [64] tested that the SHG intensity of WS_2_ spiral structure increased exponentially with the number of layers. Through spherical aberration corrected transmission electron microscope observation, it was found that the opposite SHG characterization result was due to the twisted spiral structure breaking the inversion symmetry. The twist angle of the spiral structure was measured by HRTEM to be 5 degrees and confirmed by the polarization test of the SHG signal.

In addition to the commonly used SHG to determine crystal orientation, for materials that do not have an SHG response, THG is also used to detect the crystal axis orientation of the crystal. Wu et al. [51] used a THG microscope to quickly and extensively characterize the exfoliated BP. The anisotropy of the THG signal can be used to quickly characterize the BP’s crystal axis orientation (Figure 6e). The dependence of the strength of THG on the number of layers is also studied.

With the continuous development of the technology of using NLO to study the orientation of the crystal axis, researchers began to further study other related phenomena (such as crystal phases, defects), which provides a powerful characterization tool for the application of 2D materials.

#### 3.2.2. Crystal Phase

The crystal phase is an intrinsic property of 2D materials. Different crystal phase transitions are a complex process that may involve interactions between lattices, layers, and interactions between strongly correlated electrons, which is also a process of discovering new materials. Meanwhile, the preparation of materials with different crystal phases has also become a demand. To date, many methods have been developed to achieve crystal phase control in 2D materials [65]. Existing methods, such Raman spectroscopy, XRS, and TEM are commonly used to characterize the crystal phase [66,67], but due to the lack of real-time monitoring capabilities of these characterization methods, the phase transitions have not been studied deeply. Nonlinear optical microscopy has recently attracted more and more attention for characterization of the crystal phase and the process of phase transition due to its advantages of optical, non-invasive, and real-time imaging capabilities.

The existence of different crystal phases greatly enriches the characteristics of 2D materials. For example, for transition metal dichalcogenides (TMDs), the semiconducting properties of the 2H phase are very suitable for electronic and optoelectronic devices [68,69], and the 1T and 1T’ phases represent semi-metal properties, which are beneficial for the preparation of devices with magnetoresistance and superconductivity [70]. Taking MoTe_2_ as an example, research groups have used different methods to induce samples with known phases. For example, laser ablation is used as a heat treatment method. Yu et al. [44] used this method to induce the phase transition of a few layers of MoTe_2_. As shown in Figure 7a–d, polar plots change from the original six-petal pattern to a two-petal pattern in the induced area, which shows that the crystal phase changes from the original 2H phase with semiconductor characteristics to the 1T’ phase with metal characteristics after laser ablation, and the point group symmetry changes from $C_{3h}^{1}$ to $C_{S}^{1}$. In addition to the above methods, electrostatic gating is a feasible solution to control the phase transition in monolayer TMDs. Given that there is an inversion symmetry in the monolayer 1T’ phase–MoTe_2_ that is absent in the 2H phase, SHG microscope can be a sensitive probe to distinguish the two phases of monolayer MoTe_2_. The process of phase transition of monolayer MoTe_2_ induced by an electrostatic gate is roughly as follows: the gate bias is adjusted under the forward bias voltage of up to 4 V. When the gate bias is adjusted upward from 0 to 2.2 V, the SHG intensity will gradually increase, which is due to the change of the absorption distribution by doping in the 2H phase. When the bias is further increased from 2.2 V to 3.6 V, the SHG intensity drops sharply, whose dropping degree is more than one order of magnitude. The absorption change caused by doping in the 2H phase is excluded as the main influencing factor, so the gradual disappearance of this SHG signal is attributed to the restoration of inversion symmetry caused by the phase transition from the 2H phase to the 1T’ phase. Furthermore, the polarization-resolved SHG is used to verify the reversibility of the crystal orientation. At original 0V, the polar plot is a six-petal pattern. After gate scanning of a full loop, the polar plot coincides with the original. Such a reversibility of crystal orientation is important for phase-transition devices with repetitive cycles [71].

In addition to MoTe_2_, the crystal phase of the layered group III–VI semiconductors has also been studied by SHG [43]. It is theoretically proved that when hydrostatic pressure is applied to the InSe material, a transition of the crystal structure can be achieved. The evolution of the polar plots revealed by SHG microscopy shows a symmetrical transition from 3-fold rotational symmetry to mirror symmetry, and when the pressure is released, this structural change is restored to the original state. During the change of pressure, the SHG intensity remains at the same order of magnitude, and it does not suddenly disappear, indicating that the phase transition does not occur when the structural symmetry changes [72]. Alloying 2D materials is a novel way to tune the SHG intensity. It is revealed by the polarization-resolved SHG that this method does not cause a phase transition of crystal, which has been illustrated both in InSe and TMDs [43,73].

Different synthetic methods are essential for synthesizing 2D materials with different crystal phases, and the rapid identification of crystal phases is beneficial to optimize the preparation process. For example, the cooling step plays a crucial role in the synthesis of 2H and 1T’ MoTe_2_. As shown in Figure 7e, it is the polar plot of SHG intensity of a three-layer 1T’ MoTe_2_ synthesized under rapid quenching, which can be fitted to ${\left|\alpha cos^{2}\theta +\beta sin^{2}\theta \right|}^{2}$. On the other hand, Figure 7f shows the polar plot of SHG intensity of a 2H MoTe_2_ synthesized under a slower cooling process, which can be modeled to $cos^{2}\left(2\theta +\theta_{0}\right)$ [74]. Zeng et al. [75] adopted a similar method to obtain WS_2_ and WSe_2_ of 3R and 2H phases. The process difference is that hexagonal 2H is obtained at a higher deposition temperature. With the gradual optimization of the growth process and the understanding of the material itself, the difference in the dependence of the number of layers, and the SHG intensity, and the shape of the synthetic material can be used as the basis for identifying the crystal phase (Figure 7g,h).

To a certain degree, SHG microscope has been applied in characterizing crystal phase as a good real-time monitoring method, but the related research is limited. It has been proven that the SHG microscope is an effective method to detect the symmetry of the crystal structure, so it is also worth exploring further from the aspect of detecting the phase transition [57,76]. In addition to the SHG microscope, other nonlinear techniques such as SFG and THG microscope have also received extensive attention [17,53], so deep research can also be done from the aspect of detecting a phase transition in the future.

### 3.3. Defects

2D materials have many excellent physical and chemical properties in a perfect crystal form, but it is easy to introduce undesirable structural defects during the growth process, which affects their performance. At the same time, it is also possible to adjust the electronic, optical, vibration, chemical, and other characteristics by introducing artificial defects in 2D materials, which can be very beneficial to the preparation of new optoelectronic devices. Therefore, the optical characterization of defects in a 2D material is beneficial to the preparation of high-quality ultrathin materials and new devices [14,77]. As we all know, 2D materials are infinitely large in the plane and have only a few atomic layers perpendicular to the plane. Therefore, the defects of 2D materials can be classified according to dimensions as zero-dimensional defects (vacancy, doping, etc.); one-dimensional defects (grain boundaries, edges, in-plane heterostructure, etc.); and 2D defects (stacking between different layers, wrinkling, folding, scrolling, etc.) [77].

Different techniques have been used to characterize various types of defects in 2D materials to assess crystal quality. For example, TEM can directly solve the atomic details of defects, but TEM is time-consuming, and the size of the TEM characterization area is limited to nanoscale [78]. Photoluminescence (PL) spectroscopy imaging can also be used to visualize grain boundaries (GBs) in materials. However, when the rotation of the crystal axis between neighboring grains is small, PL imaging is not very sensitive to GBs. Raman spectroscopy is also a relatively mature method, but similar to the PL microscope, it is a very slow measurement technique and cannot image in real time [79]. We have mentioned earlier that nonlinear optical microscopy has been extensively and deeply researched in characterizing the crystal structure. Since defects can cause changes in the crystal structure, they can also be characterized by the nonlinear optical microscopy.

#### 3.3.1. Zero-Dimensional Defects

The most common zero-dimensional (0D) defect during the process of sample synthesis is vacancy. By adjusting the number of vacancies to change the photocatalytic H_2_ evolution rate of the material [80], nonlinear optical properties [81], and other physical and chemical properties is a typical measure for defect control. From Figure 8a–c, it can be seen that as the N vacancies increase, that is, defects gradually increase, the TPEF signal weakens and the CARS signal increases. Due to the larger defect, the surface of defective g-C_3_N_4_ is of a larger roughness, so the SHG signals greatly increase [82]. For example, during the synthetic process of TMDs, for the high sulfur vacancy defect region of the monolayer WS_2_ grown by chemical vapor deposition (CVD), the SHG signals experience a magnitude enhancement of two orders due to the existence of mid-gap states (as shown in Figure 8d) [83]. This all proves that the nonlinear signals will change according to the density of defects in the material, which shows that the nonlinear signals can be used to distinguish defective areas with higher and lower density. Doped 2D materials, such as AuCl_3_-doped MoSe_2_ [84], alloyed MoS_2(1−x)_Se_2x_, InSe_0.9_Te_0.1_, InSe_0.9_S_0.1_, and InSe_0.8_S_0.2_ are also characterized by SHG [43,73,85,86]. Polarization-resolved SHG reveals that alloying can tune its second-order nonlinear response, but the point group of 2D materials will not change, which means that the structural symmetry does not change. As shown in Figure 8f–h, the polar plots of pure InSe and alloyed ε-InSe samples all show 3-fold rotational symmetry [43].

#### 3.3.2. One-Dimensional Defects

One-dimensional (1D) defects in 2D materials are sometimes extremely destructive because they directly hinder the transfer of charge, spin, or heat [78]. The grain boundary is a common 1D defect. The appearance of grain boundaries has been observed in many 2D materials, such as MoSe_2_, MoS_2_, WS_2_, and BP [53,78,79,87,88], and these 2D materials are synthesized by different methods. Figure 9a shows MoS_2_ grown by a solid-phase sulfurization strategy, and the area indicated by the yellow arrow is the grain boundary. It can be seen that the SHG at the grain boundary is strongly suppressed [88]. The suppressed SHG at the grain boundaries is attributed to the destructive interference and annihilation of the nonlinear waves generated from the adjacent grains with opposite orientation [59,88]. Since SHG and SFG both belong to the second-order nonlinear optical effect, SFG microscopy can also be used as a probe to identify grain boundaries, and the observed phenomenon is similar to SHG (see Figure 9b) [88]. The twin boundary is the simplest kind of grain boundary. The mirror twin boundary of the butterfly is clearly displayed by the SHG microscopy, as shown by the arrow in the right panel of Figure 9c. The SHG microscopy shows a strong contrast between the two wings, indicating that it consists of two mirror domains, which cannot be recognized by optical microscopies (as shown in the left panel of Figure 9c) [62]. In order to further enhance the contrast of 1D defects in 2D materials, a dark field characterization technique combining SHG microscopy and spatial filtering is developed. A 1D defect contrast factor is defined to evaluate the bright field (BF) SHG and dark field (DF) SHG microscopy, and it is found that the contrast factor of DF-SHG is three times that of BF-SHG when characterizing the grain boundary. Obvious difference regarding contrast can be clearly seen through the white line of Figure 9d. The advantages of DF conditions are also demonstrated by the imaging of MoSe_2_, MoS_2_, and WS_2_ samples under DF conditions [78]. SHG microscopy can only be used to characterize the grain boundaries of non-centrosymmetric materials. However, THG signals are not limited by structural symmetry. At the same time, as shown in Figure 9e, the grain boundaries of samples treated by chemical reagents show extremely high contrast under the THG microscopy, which could not be recognized by SHG imaging [79].

The most prominent defect in the TMD flakes is namely the edge, and the electronic structure change of the 2D crystal edge leads to a strong resonant nonlinear optical susceptibility, which enables direct nonlinear optical imaging of the atomic edges and the boundaries of the 2D material [78]. At the same time, because the synthetic materials follow the principle of low energy edges, the synthesized TMDs’ single crystal islands are usually triangles with very sharp edges or deformed hexagonal shapes under low sulfur vapor pressure [77].

Lateral hetero-interfaces are another form of 1D defects. An in-plane heterojunction interface such as WS_2_–MoS_2_, WSe_2_–MoSe_2_, WSe_2_–MoS_2_, MoS_2_–MoSe_2_, and MoS_2_–graphene synthesized by CVD [63,89,90] can also act as a p-n junction. Using polarization-resolved SHG to characterize the lateral heterostructures WS_2_−MoS_2_ and WSe_2_−MoSe_2_ generated by the CVD method, it is found that the zigzag direction of the lattice is selected for the one-dimensional interface [63].

##### 3.3.3. D Defects

2D defects can be also understood as surface defects. The Van der Waals interface has a great influence on the electronic and optical properties of multilayer 2D materials, so the interface related to the stacking and layer orientation can be regarded as 2D defects [77]. Ideally, a TMD with an even layer stacking is a 2H (Bernal) structure and does not generate a SHG signal. When a typical CVD method is used to grow a bilayer WS_2_ sample, the upper layer tends to nucleate on the grain boundaries of the first polycrystalline layer, which may result in the structural dislocation of the bilayer WS_2_ sample and not an ideal 2H structure. Due to the absence of inversion symmetry, this undesirable structure can produce unexpectedly strong SHG responses (see Figure 10a) [91]. The unusual nonlinear optical response caused by stacking is also found in some other 2D materials such as MoS_2_ (see Figure 10b) [87], h-BN [92], graphene [93], and GeSe [94]. The intensities of the SHG signal are closely related to the stacking angle and configuration. According to the difference of the stacking angle and the configuration, the different SHG intensities are displayed in the stacking area (see Figure 10c) [62], and stacking faults may result in non-rotationally symmetric polar plots of SHG [92].

Ripples and folding can introduce strain into 2D materials, which affects their electronic properties and can also be regarded as 2D defects. Through CARS, SHG, and TPEF microscopy, the quality of the graphene grown on the unpolished copper film is revealed, and ripples and streaks can be clearly observed (see Figure 10e–g). At the same time, it can be seen that CARS microscopy is superior to SHG and TPEF microscopy in characterizing morphology, which is due to the vibration resolution characteristics [95].

Second-order nonlinear optics, especially SHG, have been deeply studied in characterizing defects, but there are relatively few researches on third-order nonlinear characteristics. According to previous research, the third-order nonlinear signal will not disappear when the number of layers changes, so we have the reason to believe that third-order nonlinear optics may also have great potential for characterizing defects.

### 3.4. Strain and Chemical Dynamics

#### 3.4.1. Strain

2D materials are a relatively new class of atomically thin materials with emerging characteristics that are ideal for next-generation ultrathin semiconductor devices. Mechanical strain energy can strongly interfere with the energy band structure of these materials, creating a significant possibility of using mechanical deformation to adjust their electronic and photonic properties. At present, the characterization of strain in 2D materials is mainly carried out by electron/neutron microscopy, photoluminescence spectroscopy, and Raman spectroscopy, but these methods cannot detect the in-plane strain direction, and the detection speed is relatively slow, especially when performing Raman imaging. Since some nonlinear signals of 2D materials are sensitive to material strain, NLO detection methods with the advantage of rapid detection can be used.

The 2D material is usually transferred to the substrate so that strain can be introduced into the 2D material by controlling the deformation of the bulk substrate. By integrating a single layer of molybdenum disulfide on a 1D TiO_2_ nanowire, the anisotropy of optical SHG of MoS_2_ is strongly enhanced. The anisotropic SHG pattern is highly dependent on the stacking angle between the nanowire direction and the MoS_2_ crystal orientation (Figure 11a). The symmetry broken of the SHG polarization pattern and the anisotropic SHG enhancement are attributed to the lattice deformation induced by one-dimensional nanowires, which breaks the 3-fold crystal symmetry of MoS_2_. This indicates that SHG is very sensitive to the strain amplitude in 2D TMDs [96]. The continuous and reversible evolution of structural symmetry can be monitored in situ using polarization-resolved SHG spectroscopy. As the pressure changes, a reconfigurable symmetric transition of SHG mode from triple rotational symmetry to mirror symmetry is observed experimentally in layered InSe samples (Figure 11b). This opens up a new way to manipulate the potential application of the crystal symmetry of 2D materials, and it also proves the sensitivity of SHG to the detection of pressure changes [72].

Lukas Menneld et al. [97] experimentally determined the second-order nonlinear photoelastic tensor of MoS_2_, MoSe_2_, WS_2_, and WSe_2_ at the excitation wavelength of 800 nm. Even under small strains, the SHG response will change greatly. Liang et al. [98] reported a novel method to monitor 2D MoSe_2_ generated by a polarization-dependent optical SHG. The magnitude of strain can be evaluated in a sensitive manner by the SHG intensity (Figure 11c), and this method can detect the relative change of the strain from 1% to 49%. The strain direction can be directly indicated by the evolution of the polarization-dependent SHG diagram. Recently, Liang et al. [99] reported a general method to measure the full strain tensor in any 2D crystal material through polarization-dependent third harmonic generation, which makes up for the shortcoming of the second harmonic only used for non-centrosymmetric materials.

Compared with their bulk materials or conventional electronic materials, 2D materials are inherently able to withstand greater mechanical strains, so the strain engineering of 2D materials is more noticeable. In the future, NLO methods have great potential for the characterization of material strain.

#### 3.4.2. Chemical Dynamics

The oxide semiconductor heterostructure is the core of electrons and optoelectronics (such as field-effect transistors, FET). Therefore, the growth of high-quality oxide films on semiconductors is critical to the development of electronic devices. The simplest method of growing oxides on semiconductor materials is surface thermal oxidation, so laser oxidation has also become a research hotspot. At present, the commonly used characterization methods for detecting the oxidation effect of 2D materials include Raman spectroscopy, photoluminescence spectroscopy, and AFM. Since the oxidation process has a huge impact on the performance of TMD-based devices, it is important to study the oxidation process in TMDs. NLO has played an important role in studying the dynamics of 2D materials due to its advantages of fast imaging speed, large detection area, and no damage, especially the application of real-time monitoring.

The combined SHG-FWM was used for structural and electronic imaging to monitor the oxidation kinetics of MoS_2_ and its spatiotemporal evolution. As the annealing time increased, the SHG signal continued to increase, and the FWM signal gradually decreased (Figure 12a). This indicates that the central area of the sample gradually became a complete MoS_2_ monolayer [17]. Graphene oxide (GO) is a functionalized graphene with oxidized groups (mainly epoxy and hydroxyl groups). Due to its large surface area, excellent physical and chemical properties, and easy composition with other materials through surface functional groups, GO has attracted a lot of attention in the past decade. For the photo-oxidation of graphene, a wide-field FWM microscope can be used to monitor the GO pattern in real time (Figure 12b), and the changes caused by photo-oxidation are displayed as dark areas in the FWM image. Compared to commonly used Raman imaging, this method provides a sensitive, non-destructive way for the rapid large-area characterization of graphene [100].

Different nonlinear detection methods have different responses to the chemical kinetic detection of 2D materials. In the future, we believe that combined multi-modal NLO imaging is expected to be a powerful non-invasive tool for monitoring chemical reactions in 2D materials.

### 3.5. Chemical Specificity and Ultrafast Dynamics of Excitons and Phonons

#### 3.5.1. Chemical Specificity

Raman spectroscopy is one of the commonly used characterization methods. The Raman spectrum of each 2D material has its specific Raman fingerprint peaks, such as the D peak (defect peak) and the G and 2D peaks of graphene (its peak position, half-height width, and the ratio of the two will change with the change of the number of graphene layers). Through these characteristic peaks of 2D materials, we can not only distinguish the material itself but also obtain its layer number, defects, and even crystal structure information through the peak position, peak intensity, and half-height width of the characteristic peak, that is to say, the fingerprint peak of 2DLM carries its chemical specificity information. However, conventional Raman spectroscopy with weak signal intensity always has the issue of the time-consuming measurement, which prompts us to find new characterization methods. CRS (CARS and SRS) is a kind of coherent Raman spectroscopy that can effectively make up for these shortcomings; its advantages and disadvantages are compared as shown in the Table 1 below [26]. The application of CRS in the field of biology has been very successful, but according to the current research, the application in the field of 2DLM is still very limited. The application of CARS is limited by the influence of its non-resonant background and system sensitivity, because many Raman characteristic peaks of 2DLM are mostly distributed in the region of small wave numbers. However, SRS is not limited to these restrictions in theory, which makes it a promising method for 2D material characterization.

The research on graphene CARS spectral characterization and scanning image are relatively mature. For example, in the article of Galyna Dovbeshko [101], the CARS and Raman spectra of various carbon materials were tested, such as GO, graphene nanoplatelets (GNPs), and highly oriented pyrolytic graphite (HOPG). The CARS peaks corresponding to the D and G peaks in the Raman spectra of these materials have been found significant shifts (where the D peak is shifted less than the G peak), and new GCARS (corresponding to the G peak in the Raman spectrum) peaks appear, as shown in Figure 13a–c. The CARS of GNPs does not have a 2D mode at 2600 cm^−1^, and a second-order enhancement mode appears at 2960 cm^−1^, as shown in Figure 13d. The comparison between the CARS of these materials and the characteristic peaks in their Raman spectrum is shown in Table 2. Similar to Raman spectroscopy, graphene’s CARS spectrum also shows D and G peaks carrying information such as defects and layer numbers. The characteristics of graphene samples can be obtained by analyzing its CARS spectrum. In addition to the CARS spectrum of graphene, the CARS signal at a fixed Raman wave number of graphene is used to scan and image the micro area, as shown in Figure 13e, which is the CARS scanning image at 1330 cm^−1^ (D peak) Raman wave number [95]. Compared with SHG and TPEF, the specific CARS peak scanning image is clearer and shows more defect details.

Figure 13f,g show the image and spectrum of h-BN. Figure 13f shows the image of h-BN under optical microscope, SRS, CARS, and SHG respectively. It can be seen that the scanning image effect of h-BN under SRS is closest to optical microscope, and its intensity and contrast are much better than those of optical microscope. The contrast and the signal-to-noise ratio of the CARS scanning image is the worst, because CARS is affected by nonresonant background (the methods of signal enhancement, time resolution, and polarization detection can be considered to suppress the effect of non-resonant background) as an optical parametric process [26]. The strength and weakness of the SHG signal is due to its dependence on the broken central inversion symmetry of the material. Figure 13g shows the Raman and SRS spectral information extracted at the same position of the same h-BN material. Compared with the Raman spectrum, the signal-to-noise ratio of SRS is much higher [48]. As far as CARS and SRS are concerned, SRS is not affected by the non-resonant background, so the signal-to-noise ratio is better than CARS, while its system is more complicated. If possible, we recommend using SRS to perform the spectral characterization of 2D materials.

The selection rules in Raman spectroscopy cannot be simply applied to CRS spectroscopy because of the big differences between them, so machine learning methods are proposed to help us discover and apply the rules in CRS spectroscopy and better characterize and analyze the number of layers, defects, and even electrical characteristics of 2D materials. We start with the classification of graphene layers. As shown in Figure 13h, the CARS spectra of the two and three layers of graphene cannot be distinguished by the spectral image by the human eye, but we use the support vector machine (SVM) and Gaussian process classifier (GPC) to learn and classify the wide-spectrum CARS of graphene. The dataset includes one to four layers of graphene samples, and each layer has about two to four samples. A total of 368 sets of graphene CARS spectra were collected, of which 250 were used as the training set, and the remaining 118 sets were used as the test set. Two methods are used to classify the graphene layers, and the accuracy is as high as 95.8% and 96.6%, respectively. Figure 13i shows the classification results of SVM.

So far, there has been relatively little research on the CRS spectra of 2D materials, and the relationship between CRS and the physical and chemical properties of the materials is not very clear, so CRS spectra cannot be quickly invested in the characterization of 2D materials. For example, although the CARS peaks of GNPs have been fully detected in the article of Galyna Dovbeshko et al. [101], because there is a big difference between the CARS and Raman spectra, we still do not know the relationship between the change of the intensity, peak position, and half-width information of these peaks with the number of layers and defects, etc. However, because of this, CRS spectroscopy has great development potential in the characterization of 2D materials, and more efforts are needed to provide an in-depth study in this unexplored area.

In addition to CRS, vibrational SFG also has chemical specificity. When one of the photons is adjusted to match the vibration transition, vibrational SFG will be enhanced by resonance to detect the vibration mode. Such a process involves infrared absorption and anti-Stokes Raman transition processes, so the molecular vibration mode must have both IR and Raman activity to appear in the vibrational SFG spectrum [102]. Since vibrational SFG has less research work on 2D materials, it will not be discussed in detail.

#### 3.5.2. Ultrafast Dynamics of Excitons and Phonons

Exciton is a basic elementary excitation in solid, which is an electron hole pair bound by Coulomb interaction. After the material absorbs a photon, the electron transitions from the valence band to the conduction band, but the electron is still associated with the hole in the valence band due to Coulomb action. Excitons are important for describing the optical properties of materials, and exciton effects have important effects on physical processes such as photoluminescence and optical nonlinearity [103]. Generally, the research on the exciton effect of 2D materials mainly uses exciton resonance to improve the conversion rate of the nonlinear response. Conversely, NLO response can also be used to characterize the related dynamics of excitons. In the study of Bose–Einstein condensation, a large part of the content is related to exciton dynamics, such as exciton complex luminescence dynamics, exciton relaxation dynamics, and exciton diffusion dynamics. It is of great significance to explore the related dynamics of excitons.

For a single 2D material, the coherence and population dynamics of exciton transitions in the MoSe_2_ monolayer can be measured by using an FWM microscopy, which reveals that their phase shift time $\tau_{2}$ and radiation lifetime $\tau_{1}$ are in the sub-picosecond (ps) range (close to $\tau_{2}$ = 2$\tau_{1}$). This results in a radiation-limited phase shift at a temperature of 6K. In addition, the phase-shifting mechanism is elucidated by changing the temperature and detecting the positions of different local disturbances on the thin plate [104]. For heterojunction materials, this research method is also applicable. In the MoSe_2_ and WSe_2_ monolayers encapsulated between hexagonal boron nitride thin films, the variation of excitons’ uniform spreading with environmental factors (i.e., temperature and exciton density) is measured. By using FWM imaging, it is found that the nonlinear excited absorption of excitons and their coherent coupling can be observed at such a position (Figure 14a) [103]. Similar methods are also used in related research [105,106,107].

Phonons are the quantization of lattice vibrations. The introduction of the concept of phonons makes many complex physical problems easy to handle. Phonons not only have an important influence on the thermal properties of crystals such as specific heat, thermal expansion, and thermal conduction, but they also are closely related to the electrical, optical, and dielectric properties of the crystal [108]. Using the interaction of three laser pulses with different wavelengths, a time delay (Figure 14b) between the two pulses is set to generate a CARS signal with time resolution. The three-pulse CARS signal shows that the G-mode phonon lifetime of graphene is 325 ± 50 fs; it decays exponentially due to the phase shift of the G mode. The time-resolved CARS method provides a feasible research tool for detecting the phonon dynamics of 2D materials.

## 4. Discussion

### 4.1. The Potential of NLO Characterization of 2D Materials

In the development of 2D materials, characterization has always been a crucial topic from the recognition of 2D materials to the application of 2D materials. Although a variety of characterization methods have become important research tools for researchers, there is no one-size-fits-all characterization technique that can solve all characterization problems. Fortunately, NLO characterization methods provide us with a good idea to simplify this problem.

In part 3, we introduce in detail the use of NLO methods to characterize various physical and chemical properties of 2D materials. For 2D materials with few layers, the thickness can be judged accurately by selecting or combining suitable NLO characterization methods. Especially at a large area condition, NLO provides a more rapid and intuitive method. Usually, SHG and FWM are used for characterization, where FWM can theoretically be generated from all 2D materials, and SHG is suitable for materials with broken inversion symmetry. For accurately characterizing the number of material layers, it is recommended to combine SHG and FWM or even more NLO methods to perform multi-modal characterization. THG is often used to characterize its number of layers for the materials with inversion symmetry such as black phosphorus, and its FWM signal conversion efficiency is much lower than that of the THG signal.

In addition, the use of NLO methods to study the crystal structure of materials has become a commonly used method, especially through SHG, whose signal is very sensitive to the azimuth of the crystal axis orientation relative to the polarization direction of incident light. The odd number layers of the material such as MoS_2_ and WS_2_ whose single layer is inversion symmetry broken, have a strong SHG response, whereas the even number layers of the material such as MoTe_2_, WTe_2_, ReS_2_, and ReSe_2_, whose single layer has inversion symmetry, have a strong SHG response due to the symmetry breaking caused by stacking. The crystal phase of 2D materials is indispensable for structural engineering and characteristic modulation, which has attracted much attention worldwide. Rapid identification of the crystal phase is crucial to achieving the above goals. In Section 3.2.2, the phase transition induction of 2D materials by laser ablation, electrostatic gating, and other methods is summarized, and the process is demonstrated by SHG microscopy. This part also summarizes the use of SHG microscopy to identify 2D materials of different crystal phases generated by different synthetics methods, which is essential for improving the preparation process. Defect characterization is the key to understanding the material structure and enabling specific applications. Defects are divided into 0D defects, 1D defects, and 2D defects according to dimensions, and the characterization of various defects is summarized by nonlinear means in Section 3.3. For the 0D defects of 2D materials, such as vacancies, doping, SHG, TPEF, and CARS are commonly used. For one-dimensional defects of 2D materials (such as TMDs, BP), such as grain boundaries, edges, and lateral heterogeneity, common nonlinear characterization methods include SHG and THG. For 2D defects such as stacking errors and wrinkles that may exist in 2D materials such as TMDs, h-BN, graphene, and GeSe, the SHG, CARS, and TPEF microscopies are applied. Interestingly, in the process of characterizing defects, the imperfections of these structures are not completely harmful. In some cases, defects will bring benefits to certain material properties, enhance device performance, and achieve unprecedented functions.

Similarly, mechanical strain can also strongly interfere with the crystal structure of the material and significantly change the NLO response of the material, resulting in a signal that is extremely sensitive to strain. The selection of the detection method refers to the crystal orientation. In addition, CRS is chemically specific, and its feasibility has been verified on graphene CARS and hexagonal boron nitride SRS. SRS technology not only has chemical specificity, but it also has the sensitivity to the number of layers and anisotropy of other parametric processes such as second harmonic and third harmonic. In summary, although NLO does not have much research on chemical specificity currently limited by hardware technology, it is expected to form a unique and colorful research field in the future.

Based on the powerful ability of NLO to characterize 2D materials, researchers usually have a mature NLO system and then explore various properties of 2D materials. In particular, the combination of multiple modes will find extraordinary research results. NLO characterization has brought great benefits to the research of 2D materials; with the further exploration and application of this technology, we believe that NLO will show great potential in the characterization of 2D materials in the future.

### 4.2. Challenges and Opportunities of System Instrument

At present, there are some challenges in most system structures that use NLO to characterize 2D materials. For example, a system is only suitable for a certain NLO characterization method, and the function is simple. When using spectrally narrow excitation light for broadband spectrum detection, the excitation wavelength requires to be adjusted continuously, which makes the processes complicated. Not only it is very inconvenient to use and the characterization efficiency is low, but also the characterization result may be inaccurate due to the long detection time. Meanwhile, the acquisition speed is slow, and it is difficult to detect the rapid crystal phase transition in real time. This means that there are many aspects that need to be improved for the NLO characterization system of 2D materials.

Nowadays, most researchers use the single NLO characterization method. As the instrument is relatively simple, the characterization of 2D materials is incomplete in many cases, which shows that using a single characterization platform to identify key characteristics of 2D materials comprehensively remains a challenge. Therefore, multimodal NLO microscopy can be considered [109,110], which integrates different NLO characterization methods in the same instrument and can realize the measurement of SHG, THG, coherent Raman, and TPEF. In this way, the versatility and detection efficiency of the system can be improved, and it is also easier to operate. Different characterization methods can be selected according to the different properties of different materials. Multi-modal imaging technology can also be selected to comprehensively characterize the properties of the material such as surface morphology, number of layers, symmetry of the crystal structure, defects, chemical specificity, and strain. For instance, this method can be used to make up for the shortcoming that SHG cannot fully detect the number of layers due to its dependence on inversion symmetry breaking. At the same time, it can make the detection results more accurate, and it also has the ability of detecting the chemical kinetics of 2D materials and the interlayer coupling of heterojunctions [17,95].

The introduction of a supercontinuum in the instrument can sometimes simplify the experimental steps, shorten the duration, and make the results accurate. For example, in the study of excitonic resonance phenomenon of 2D TMDs by using SHG/THG detection, the characterization of an excitonic resonance phenomenon can be achieved by gradually scanning with a spectrally narrow excitation pulse in a wide spectrum. This experimental technique requires precise control of the pulse duration, pulse energy, and spatial position of the beam on the sample in each adjustment step, which is difficult to achieve in a long detection time [111]. Therefore, the supercontinuum spectroscopy can be used to overcome these shortcomings. This technique has been well used in the detection of semiconductor materials and plasmonic nanostructures [112,113]. By using supercontinuum laser pulses to generate SHG and then realize exciton resonance measurement, a SHG signal covering almost the entire visible spectrum can be generated in one measurement, and multiple excitons resonance and exciton states in the broadband SHG spectrum in the material can be measured simultaneously [114]. While implementing supercontinuum in CARS, namely combining broadband pulses and narrowband pulses, multiplexed CARS measurements (multiplex CARS) can be achieved with broadband and narrow band pulses as Stokes and pump beams, respectively. The light beam can simultaneously excite molecular vibrations of multiple energy levels of the material sample, so multiple excitation Raman modes can be detected. In addition, two or more chemical substances can be identified in one scan. This technology has been well applied in the detection of biological and polymer materials [38,115], so the technology can be used for the characterization of 2D materials, and it also has great advantages in its chemical specific detection. The measurement of multiple characteristic peaks by tuning the spectrally narrow wavelength can be achieved, but the introduction of the supercontinuum makes the operation of detection easier and able to achieve rapid spectral characterization [22,108].

The system instrument can be further improved in other aspects. For example, if you need to use the CARS system to detect the dynamics of electrons and phonons in 2D materials, the time-resolved method can be used in CARS to detect the phase movement mechanics of specific modes of electrons and phonons in 2D materials [108]. At the same time, it can be considered to improve the NLO characterization platform in the inherent performance of the instrument itself. In terms of speed, the imaging speed is improved in hardware performance by increasing the scanning speed of the galvanometer and the acquisition speed of the detector. Using multiple beams to simultaneously stimulate the signal can also increase the speed [37]. Therefore, real-time NLO signal imaging can be realized, which is beneficial to real-time monitoring of the generation and change of the crystal phase of the 2D material and the defects generated during the material generation process. In SRS, the signal-to-noise ratio can be improved by using tuned amplifiers instead of lock-in amplifiers [116].

As shown in Figure 15, the multi-modal nonlinear optical imaging system of this project team has made many improvements on the basis of predecessors. The imaging system is composed of three optical paths. The polarization state and laser power can be adjusted manually through a half-wave plate and a Glan-Laser polarizer behind the beam splitter. The optical path of one laser beam is adjusted by the optical delay line so that the two lasers can keep time synchronization, and then pass the photonic crystal fiber to produce a supercontinuum. Then, two laser beams of different wavelengths are combined by a beam splitter and expanded by the beam expanders. The 2D galvanometer realizes the laser focus scanning on the 2D plane, and a 4F system equivalently maps the deflection angle of the galvanometer to the light entrance of the objective lens. Finally, the excitation beam is accurately focused on the micro/nano structure sample of the high-precision three-dimensional nano-displacement platform through the objective lens, which can realize the CARS characterization of the material. The third beam can be used for SHG, THG, TPEF, and other NLO characterization methods. In addition, our instrument includes a time-resolved function by adding the second optical path and the first and third optical path together to form a time-resolved optical path. A delay line was added, and different beams were combined via a dichroic mirror, which can measure dynamic non-equilibrium electron distribution and relaxation in materials. The signal reflected by the nonlinear interaction between the incident laser light and the micro-nano structure sample is reflected through the dichroic mirror and received by the scientific complementary Metal-Oxide-Semiconductor detector and CCD in the imaging spectrometer to detect the spectrum and image. A bandpass filter is added between the spectrometer, CCD, and dichroic mirror, which is used to filter the excitation light to obtain the signal [108]. Compared with the optical path introduced earlier, a broadband spectrum detection function is added to detect multiple fingerprints of a sample at the same time. In addition, the time-resolved optical path is added, which can effectively analyze the dynamics inside the material. It also has the advantages of multimodal NLO imaging, and it can perform a variety of NLO imaging detection methods to satisfy different demands.

### 4.3. External Modulation of NLO Signals

The introduction of external modulation in the characterization of 2D materials mainly refers to the introduction of external energy fields (including light, electric, or magnetic fields) and external environmental regulation (substrate, temperature). It can achieve the function of regulating the nonlinear signal without destroying the structure of the tested material. On the one hand, it is mainly to improve the conversion efficiency of NLO to achieve a better detection effect on the signal. The 2D materials with a single layer and few layers are too thin and the interaction length between the light field and the material is limited to the nanoscale, which results in the low total conversion efficiency of nonlinear optics and limitation of the accurate characterization of material properties by NLO. On the other hand, the NLO response of the 2D materials can be adjusted by the external field to meet our needs for the preparation of NLO devices with specific functions.

Changing the properties of the light field can have a great effect on the enhancement of NLO signals. The exciton effect plays an important role in the NLO signal generation process. When the energy of the incident light field resonates with the exciton, the NLO signal can be significantly enhanced. 2D materials with different band gaps will have different exciton resonance wavelengths. Therefore, we can change the wavelength of the incident light, and the corresponding wavelength’s enhancement of the NLO signal will have a positive effect on the characterization of the material [58,117]. In addition to the wavelength, the polarization state can also be changed. Linearly polarized light is often used to study the anisotropic response of materials to nonlinear optical signals. However, in some scenarios, circularly polarized light can selectively excite excitons in some valleys of 2D materials [118,119], thereby obtaining a nonlinear signal response stronger than that of linearly polarized light [58]. In addition, the use of circularly polarized light to judge the chirality of the material is also worth exploring in future research.

By adjusting the external electric field, the material can produce certain nonlinear responses that would not otherwise occur. The vertical electric field causes bilayer graphene without SHG response to generate SHG signals. Under the action of the electric field, zero band gap graphene produces band gaps and destroys its lattice symmetry, which adds a new nonlinear method to the study of the properties of graphene [120]. This adds a new nonlinear method for studying the properties of graphene. In the MoS_2_ of even-numbered layers, the electric field is applied to change the crystal inversion symmetry to control and enhance the nonlinear optical process. This switchable control method provides the possibility for the realization of ultrafast optical modulators in the future [121].

In addition to the excitation wavelength, the substrate is also a crucial factor for NLO characterization. The type and thickness of the substrate have a significant influence on the strength of the nonlinear signal and the symmetry of the crystal structure characterized by the NLO signal. On the one hand, the accurate analysis of the influence of the substrate helps us choose the appropriate substrate in the experiment to achieve the modulation of the nonlinear signal. On the other hand, by excluding or considering the influence of the substrate on the characterization results of the 2D materials, the characteristics of the 2D materials can be analyzed more accurately [122].

### 4.4. Internal Modulation of NLO Signals

In the process of characterizing 2D materials, some material properties, such as the number of layers, crystal structure, defects, and strain can produce extraordinary nonlinear responses. In addition to characterizing these characteristics, these nonlinear responses can also be used to realize the preparation of nonlinear optical devices with various functional requirements. At the same time, although the current regulation technologies for these characteristics have attracted widespread attention, some technical difficulties still need to be resolved.

The number of layers and the symmetry of the crystal structure are the most basic properties of the 2D material that significantly affect the electronic band structure and lattice vibration of 2D materials. The following techniques for these two characteristics can be used to control the nonlinear response: (i) precise control of the number of layers of 2D materials grown over a large area; (ii) the precise thinning of 2D materials of known thickness in terms of atomic layer accuracy; (iii) control of the stacking method and stacking angle between different layers. Due to the limitation of technical level, the above methods are still subject to certain limitations in practical applications, but with the further development of monitoring technology and the synthetic process, the internal control of nonlinear signals through the above methods will be widely used.

The reasonable creation and utilization of defects are essential for the modulation of nonlinear signals of 2D materials. Although the treatment of defects at the atomic scale is still a challenge, we believe that with the further development of spectral characterization technology and the deepening of first-principles calculations, the following technologies for defects are worth further exploration: (i) introducing defects of a specific type and specific concentration into the desired location; (ii) migrating defects existing in the 2D material itself to a specific location; (iii) repairing the undesirable defects in the preparation of materials. Through the above methods, defects can be widely used in defect engineering to modulate nonlinear characteristics.

As an emerging technology for studying 2D materials, strain engineering can change the energy band structure of 2D materials which can affect their photoelectric properties. Although it is of great significance, there are still some problems that need to be solved: (i) precise control of the magnitude and direction of the strain to achieve the preparation of angle-dependent nonlinear optoelectronic devices; (ii) choice of strain among many strain methods according to application requirements; and (iii) evaluating correctly the various strain limitations of 2D materials to avoid the additional undesirable results on 2D materials.

In addition to the above-mentioned techniques, we can also use interface engineering, such as the preparation of specific vertical heterojunctions, in-plane heterojunctions, and composite systems formed by surface plasmon structures and 2D materials [67]. Patterning an “array structure” or “antenna structure” and other artificial microstructure technologies to improve optical nonlinearity are also available [123].

### 4.5. Machine Learning: Powerful Support for Quantitative Characterization

Although the nonlinear optical characterization methods have great potential in characterizing various characteristics of 2D materials, it is still in the qualitative description stage. For example, we can only judge the position of the thinner layer by FWM scanned images of two different areas of the same sample or determine the approximate arrangement direction of nanowires in the nanowire assembly pattern based on a polarized SHG scanned image [124]. For the accurate characterization of sample characteristics, it is necessary to combine other traditional characterization methods, which does not seem to be able to fully utilize the power of nonlinear optical characterization methods. We think of introducing machine learning methods to help us discover and summarize the laws in nonlinear optical signals and use them [125,126], hoping to achieve the goal of classifying and even quantifying the characteristics of 2D materials.

Machine learning methods can find features with varying degrees of influence on the results from a large amount of data, and they show the degree of influence in the form of weights. People can analyze the new data according to the features with relatively large weights, or they can directly input these data into the trained model to get the final result. This is undoubtedly a powerful auxiliary tool for the analysis of nonlinear signals of 2D materials. In addition to the aforementioned layer number classification judgment, we believe that at least the following points can be studied in depth. (i) Defects: 2D material defects cause changes in the crystal structure and thus affect the nonlinear signal, but in fact, there is no clear correspondence between the impact of different types and degrees of defects on different nonlinear signals, which makes it more difficult for us to understand the defects and manipulate the defects. At the same time, due to the variety of defects and the complex mechanism of interaction with the light field, it is necessary to adopt appropriate methods to establish a corresponding mechanism of classification, which can be very beneficial to achieve efficient and low-cost defect engineering. (ii) Strain, similar to the defect situation, the incomplete understanding of various strain mechanisms makes precise control of the magnitude and direction of strain still a problem. Therefore, the establishment of a comprehensive evaluation index to evaluate the impact of various strains on nonlinear characteristics is of important guiding significance for the development of strain engineering.

### 4.6. CRS: A New Highland for 2D Material Characterization Applications

The SHG technique is a powerful characterization means for 2D materials, but it still needs to improve further. For example, the wavelength of excitation light, the interface, and the substrate are all significant for atomically thin crystals, which can be further analyzed to perfect the SHG technique. However, due to limited space, we will not expand the improved method in detail. Compared to the SHG technique, CRS has chemical specificity, and compared with conventional Raman technology with chemical specificity and low signal intensity, CRS is a higher-strength material characterization method.

It is an important analysis method to judge the number of layers, crystal structure, defects, and stacking conditions by the chemical specificity of the material spectrum, and Raman spectroscopy is the most commonly used method at present. Both CRS and Raman have chemical specificity, and compared with Raman, CRS is a higher-strength material characterization method, but the research and applications of CRS in 2D materials is far less extensive than Raman spectroscopy. We hope to find the CRS peak corresponding to each 2D material such as Raman.

As mentioned earlier, the detection of CARS in the field of small wave numbers is limited, which makes many 2D materials unable to be characterized by CARS. In theory, the CARS signal is based on the nonlinear polarizability generated by the FWM electronic vibration, and it is affected by nuclear vibration under resonance, resulting in a change in the nonlinear polarizability. Furthermore, the magnitude of the change of nonlinear polarizability is affected by the frequency of nuclear vibration. When its nuclear vibration frequency is very small (that is, its Raman wave number is small), its nonlinear polarizability change is relatively small, that is to say, the non-coherent part (nonresonant background) of the generated signal is much larger than the coherent part (CARS), which makes it difficult to detect the CARS signal [26]. For graphene with a zero band gap, since the electronic excited states in the CARS process are all real existence states, there is continuous electronic resonance inside. When the frequency difference of the pump and probe light is equal to the intrinsic vibration frequency of its atomic nucleus, an anti-resonance phenomenon appears at the characteristic peak due to nuclear resonance [127]. A higher configuration of hardware is also required, such as the stability of the photonic crystal fiber spectrum broadening, the dichroic capability of the dichroic mirror, and the sensitivity of the detector. It also needs to better suppress the nonresonant background and enhance the resonance signal to make CARS more widely used in the characterization of 2D materials.

Compared with CARS, SRS does not have the problem of nonresonant background, because it is only related to the imaginary part (the part is not affected by nuclear vibration) in the third-order nonlinear polarizability. On the other hand, the characteristic peak of CARS has changed greatly compared with Raman because of the nonresonant background [101], which actually brings a certain obstacle to the application and promotion of CARS in the field of 2D materials. According to the literature [48], we speculate that SRS has little change in characteristic peaks and peak positions compared with Raman, which brings us convenience in studying the SRS spectra of 2D materials. In theory, SRS can realize signal light detection from visible light to near infrared band (380 nm–3 um) [26], which covers almost all Raman characteristic peaks of existing 2D materials, so we believe that SRS has good application prospects in 2D material characterization. However, at present, the application of SRS in the field of 2D materials is not popular. It may result from the improved performance of various traditional characterization instruments and the complicated construction of the SRS system itself. Therefore, SRS is also facing opportunities and challenges in 2D material characterization. More experts and scholars are required to invest more research in this area.

## 5. Conclusions

Among the 2D material characterization methods, NLO methods have become effective means to explore the physical and chemical properties of materials because of their unique multi-modal characterization. In this review, we aim to provide guidance for researchers to study the characterization of 2D materials using NLO responses, and we choose the best characterization method according to the material’s properties. Expanded from the number of layers, the crystal orientation, crystal phase, defects, strain, chemical specificity, chemical dynamics, and ultrafast dynamics of excitons and phonons of 2D materials, NLO has demonstrated excellent performance as a powerful tool in characterizing 2D materials.

NLO characterization has many advantages including in situ monitoring, high-speed and damage-free measurement, large area, and multi-modality characterization. Such characterization methods have great significance in quickly grasping material properties and evaluating material potential for future applications. In addition, we can also improve the spatial resolution of NLO through local field enhancement techniques such as tip-enhanced Raman scattering to improve its imaging quality. Although we also mentioned that the current system equipment is still subject to some restrictions, these problems will be solved with the continuous development of hardware/software in the future. In particular, we discussed the huge potential of machine learning and SRS technology in future characterization applications. Machine learning helps us judge fuzzy intervals, obtain information from fuzzy qualitative to hierarchical qualitative analysis, and then from qualitative to quantitative analysis. SRS overcomes the limitations of CARS technology for the detection of 2D materials, which makes it possible to replace spontaneous Raman technology and open a door toward the research field of 2D materials.

With the development of NLO for the characterization of 2D materials, we believe that there are many feasible characterization directions that await for us to explore and discover. The exploration of these characterization directions will not only help us study the properties of the 2D material itself but also have a positive impact for applications such as the fabrication of advanced nonlinear optical functional devices.

## Figures and Tables

**Figure 1 nanomaterials-10-02263-f001:**
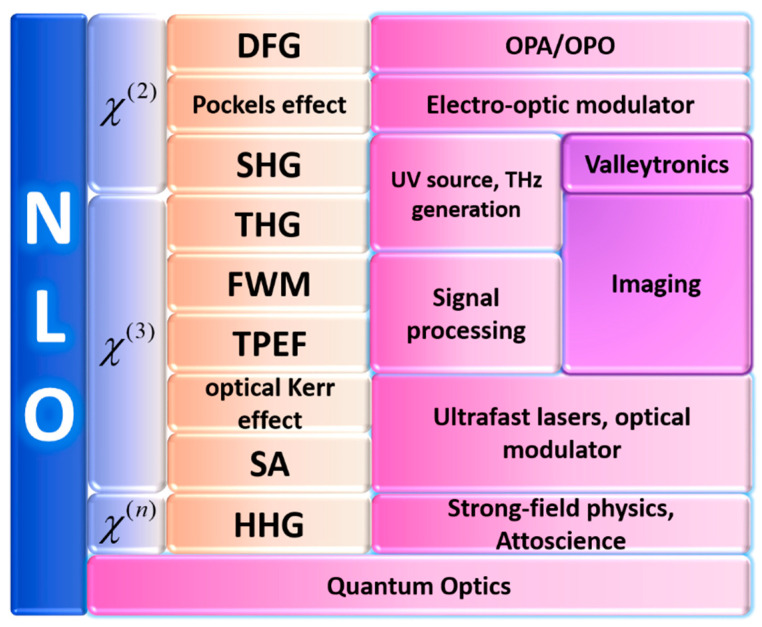
Typical nonlinear optical applications. Light blue, orange, pink and purple gradients represent different order nonlinear optics (NLO) susceptibility, nonlinear processes corresponding to different NLO susceptibility, application, and further applications respectively.

**Figure 2 nanomaterials-10-02263-f002:**
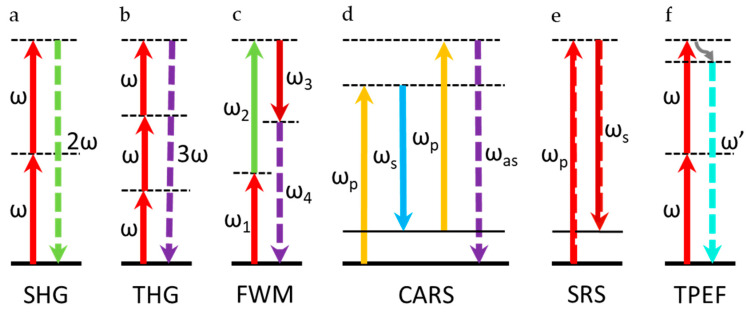
Energy diagrams of different nonlinear detection methods. In the figure, the thick black solid line represents the ground state, the thin black solid line represents the actual electron or vibration energy state, the dashed line represents the virtual state, and the solid arrow represents the incident excitation light. The arrow represents the signal light to be detected. (**a**) Second-harmonic generation (SHG) energy level diagram. (**b**) Third-harmonic generation (THG) energy diagram. (**c**) Four-wave mixing (FWM) energy diagram. (**d**) Coherent anti-Stokes Raman scattering (CARS) energy diagram. (**e**) Stimulated Raman scattering (SRS) energy diagram. (**f**) Two-photon excitation fluorescence (TPEF) energy diagram.

**Figure 3 nanomaterials-10-02263-f003:**
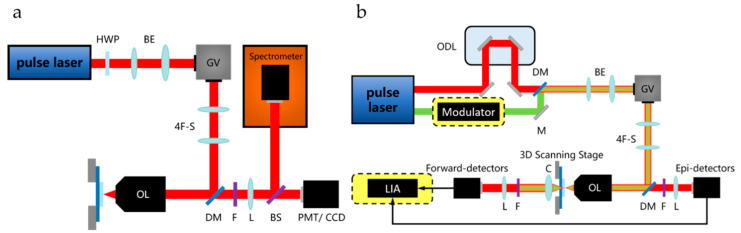
(**a**) Typical instrument of SHG/THG. (**b**) Typical instrument of CARS/SRS, the yellow dotted frame is the additional component needed for SRS system. HWP: half-wave plate; ODL: optical delay line; DM: dichroic mirror; BS: beam splitter; GV: galvanometer; 4F-S: 4F system; BE: beam expander; OL: objective lens; C: condenser; F: filter; LIA: lock- in amplifier.

**Figure 4 nanomaterials-10-02263-f004:**
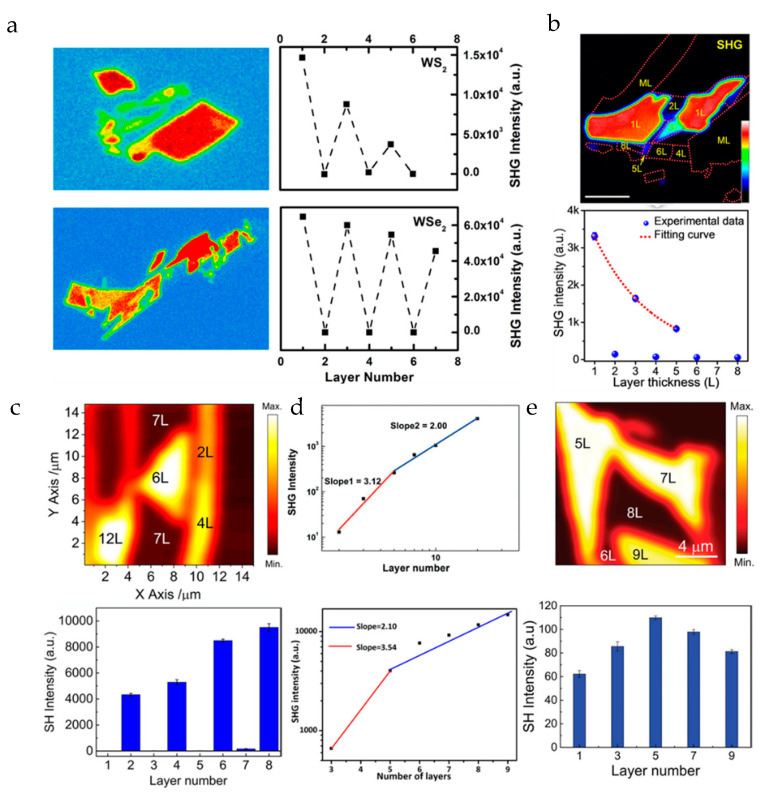
SHG scanning image of 2D materials and the relationship between the SHG signal intensity and the number of material layers: (**a**) WS_2_ and WSe_2_. Reproduced from [39], with permission from Springer Nature, 2013. (**b**) MoS_2_. Reproduced from [17], with permission from American Chemical Society, 2016. (**c**) ReS_2_. Reproduced from [41], with permission from American Chemical Society, 2018. (**d**) GaSe and InSe. Reproduced from [42,43]. with permission from John Wiley and Sons, 2014 and American Chemical Society, 2019 (**e**) MoTe_2_. Reproduced from [44]. with permission from John Wiley and Sons, 2018.

**Figure 5 nanomaterials-10-02263-f005:**
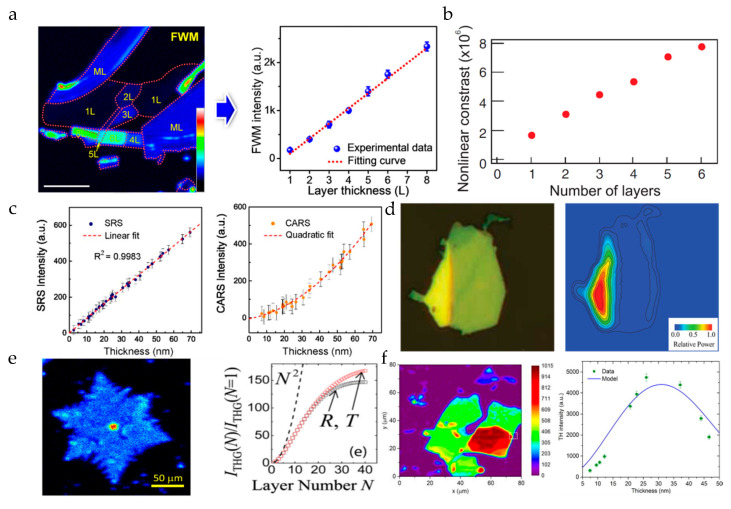
FWM scanning image of 2D materials and the relationship between the FWM signal strength and the number of material layers: (**a**) MoS_2_. Reproduced from [17], with permission from American Chemical Society, 2016. (**b**) Graphene. Reproduced from [46], with permission from American Physical Society, 2010. (**c**) The relationship between h-BN’s CRS (including SRS and CARS) and its layers. Reproduced from [48], with permission from American Chemical Society, 2019. THG scanning image of 2D materials and the relationship between its intensity and the number of layers: (**d**) MoS_2_. Reproduced from [49], with permission from American Chemical Society, 2014. (**e**) Graphene. Reproduced from [50]. (**f**) Black phosphorus. Reproduced from [51], with permission from AIP Publishing, 2016.

**Figure 6 nanomaterials-10-02263-f006:**
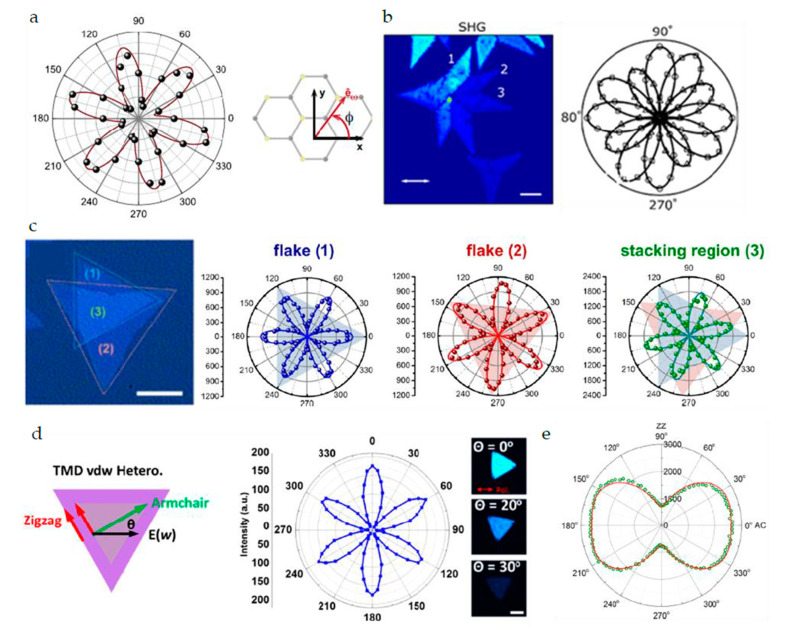
(**a**) Polar plot of the SHG intensity from monolayer MoS_2_ as a function of the sample angle. Reproduced from [57], with permission from American Physical Society, 2013. (**b**) SHG intensity depends on orientation of the flakes and laser polarization. Reproduced from [58], with permission from American Chemical Society, 2018. (**c**) The underlying crystal symmetry and crystal orientation of individual monolayer MoS_2_ flakes and the stacking regions can be further examined by the polarization-resolved SHG. Reproduced from [62], with permission from American Chemical Society, 2014. (**d**) Identification of symmetry and edge structures of lateral transition metal dichalcogenide heterostructures with SHG analysis. Reproduced from [63], with permission from American Chemical Society, 2015. (**e**) The BP flake polarization dependence of THG. Reproduced from [51], with permission from AIP Publishing, 2016.

**Figure 7 nanomaterials-10-02263-f007:**
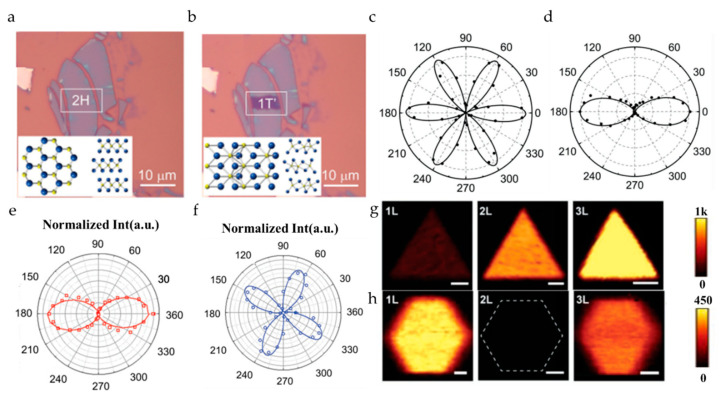
Optical microscope images of 2H-MoTe_2_ flake (**a**) before (**b**) after laser irradiation over an area marked by the white box. Insets in two panels show the lattice structures of 2H and 1T′ MoTe_2_. Reproduced from [44], with permission from John Wiley and Sons, 2018. Polar plots of the SHG intensities from the few-layer MoTe_2_ (**c**) before and (**d**) after laser irradiation Reproduced from [44], with permission from John Wiley and Sons, 2018. Polar plots of SH intensity of (**e**) 1T’and (**f**) 2H MoTe_2_, respectively. Reproduced from [74], with permission from American Chemical Society, 2017. SHG images of typical 1–3 layer (**g**) 3R, and (**h**) 2H phase WS_2_ on cover slides. The scale bars are 20 µm in the 3R phase and 10 µm in the 2H phase Reproduced from [75], with permission from John Wiley and Sons, 2019.

**Figure 8 nanomaterials-10-02263-f008:**
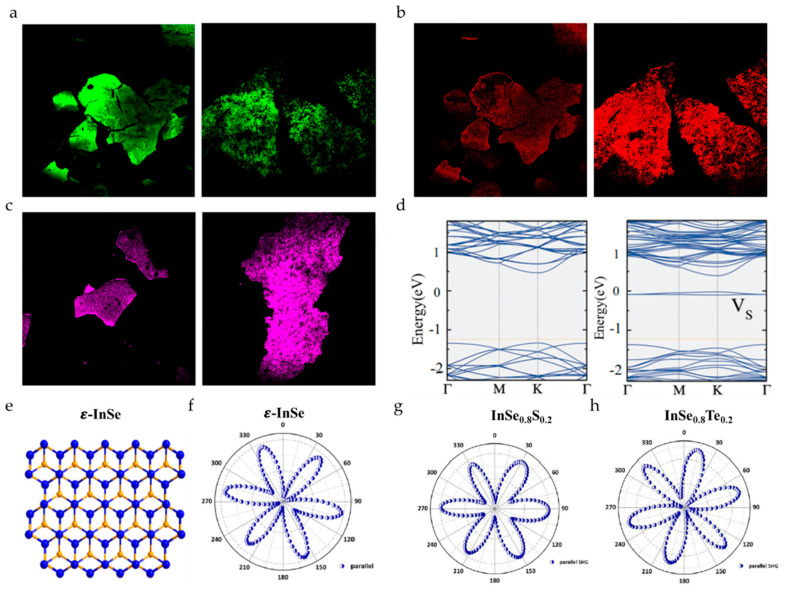
Zero-dimensional defect: (**a**) Left panel: TPEF of g-C_3_N_4_; Right panel: TPEF of defective g-C_3_N_4_. (**b**) Left panel: CARS of g-C_3_N_4_. Right panel: CARS of defective g-C_3_N_4_. (**c**) Left panel: SHG of g-C_3_N_4_. Right panel: SHG of defective g-C_3_N_4_. Reproduced from [43], with permission from American Chemical Society, 2019. (**d**) Left panel: Discrete fourier transform (DFT) calculations of electronic bands for pristine WS_2_. Right panel: DFT calculations of electronic bands for W_16_S_30_ (VS: vacancies replacing sulfur). Reproduced from [83], with permission from Elsevier, 2018. (**e**) Schematic view of ε-InSe. Polarization-resolved SH intensities of (**f**) pure InSe (**g**) InSe_0.8_S_0.2_ and (**h**) InSe_0.8_Te_0.2_ as a function of the excitation laser polarization against the crystalline lattice direction Reproduced from [43], with permission from American Chemical Society, 2019.

**Figure 9 nanomaterials-10-02263-f009:**
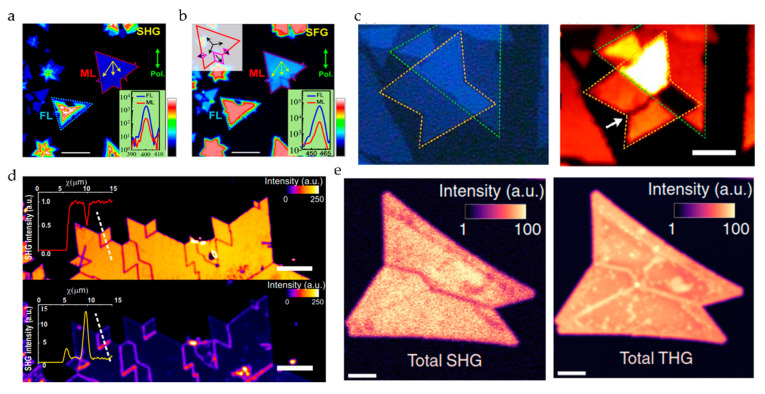
(**a**) SHG and (**b**) sum frequency generation (SFG) for the monolayer and few-layer MoS_2_, where the red and blue curves refer to the monolayer and few-layer MoS_2_, respectively. Reproduced from [88], with permission from American Chemical Society, 2018. (**c**) Left panel: An optical microscopy image for a triangular MoS_2_ monolayer overlapped with a butterfly-shaped monolayer MoS_2_. Right panel: The corresponding false color-coded SHG intensity obtained by pixel-to-pixel spatial mappings of flakes. Reproduced from [62], with permission from American Chemical Society, 2014. (**d**) Upper panel: Bright field (BF)-SHG imaging of a chemical vapor deposition (CVD) grown monolayer MoSe_2_. Lower panel: The same area under dark field. The insets depict the cross-section (dashed white line), which is used to compare the contrast of the BF-SHG and DF-SHG response on the same crystal. Reproduced from [78], with permission from American Chemical Society, 2020. (**e**) Upper panel: Experimental SHG image. Lower panel: Experimental THG image of the sample with chemical treatment. Reproduced from [78], with permission from American Chemical Society, 2020.

**Figure 10 nanomaterials-10-02263-f010:**
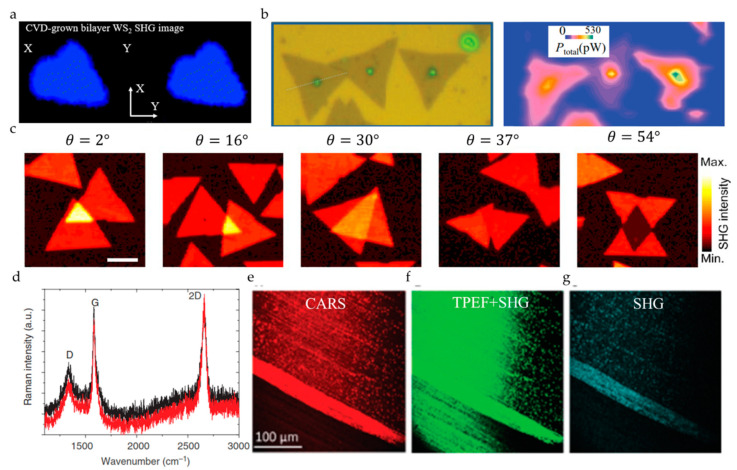
(**a**) SHG image of the CVD-grown bilayer WS_2_. Reproduced from [62], with permission from American Chemical Society, 2014. (**b**) Left panel: Optical microscopy photograph of a region of a substrate containing MoS_2_ flakes grown by CVD. Right panel: SHG microscopy photograph of the same region. Reproduced from [87], with permission from American Physical Society, 2018. (**c**) SHG intensity mapping of artificially stacked MoS_2_ bilayers with various stacking angle (θ). Reproduced from [62], with permission from American Chemical Society, 2014. (**d**) The Raman spectrum of graphene grown on unpolished Cu foils. (**e**) The CARS (Raman shift of 1360 cm^−1^). (**f**) The TPEF + SHG; (**g**) The SHG image of graphene. Reproduced from [95].

**Figure 11 nanomaterials-10-02263-f011:**
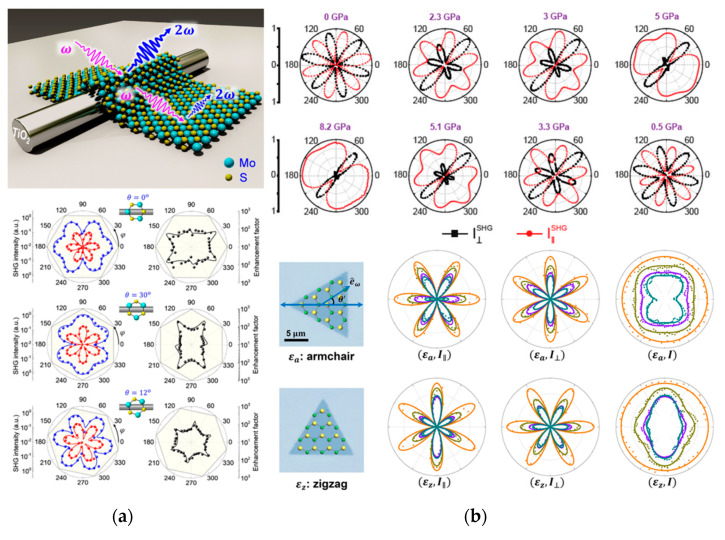
(**a**) Changes of SHG signal corresponding to different stacking angles of single layer MoS_2_ on TiO_2_ nanorods. Reproduced from [96], with permission from American Chemical Society, 2019. (**b**) Changes of symmetry of SHG signals of layered InSe under different pressures. Reproduced from [72], with permission from WILEY-VCH Verlag GmbH&Co.KGaA, Weinheim, 2019. (**c**) Pattern evolution of SHG intensity for monolayer MoSe_2_ under strain. Reproduced from [98], with permission from American Chemical Society, 2017.

**Figure 12 nanomaterials-10-02263-f012:**
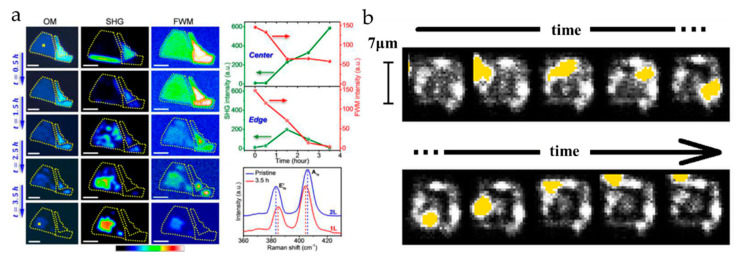
(**a**) Variation of optical, SHG, FWM images of the thermal oxidation process of an MoS_2_ sheet with different heating times. Reproduced from [17], with permission from American Chemical Society, 2016. (**b**) Time-varying image of the FWM signal of photo-oxidation graphene, where yellow is the oxidized laser beam and white is the FWM signal. Reproduced from [100], with permission from AIP Publishing, 2016.

**Figure 13 nanomaterials-10-02263-f013:**
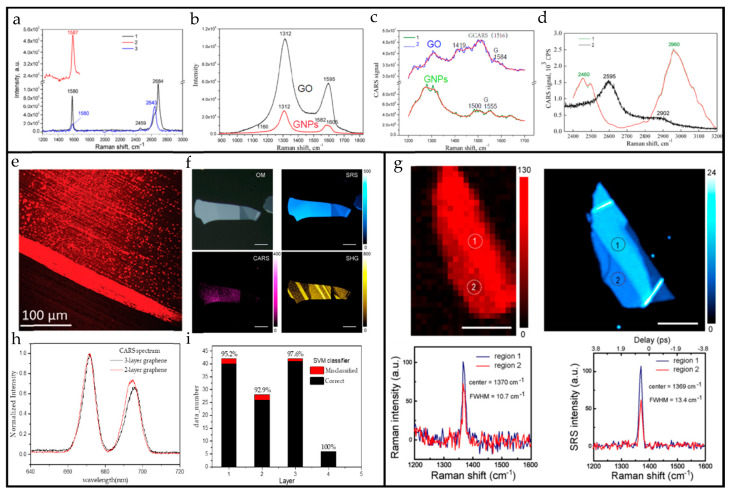
(**a**) The Raman of highly oriented pyrolytic graphite (HOPG) and single-layer graphene, and the CARS of HOPG: (1) The Raman spectrum of Highly oriented pyrolytic graphite (HOPG) (2) The CARS spectrum of HOPG, a blue shift of 7 cm^−^^1^ relative to Raman toward the large wavenumber region (3) Raman spectra of monolayer graphene on Cu substrate. (**b**) Raman spectra of graphene oxide (GO) and graphene nanoplatelets (GNPs) at 1200 cm^−1^ to 1700 cm^−1^ wavenumber. (**c**) CARS spectra of GO and GNPs. (**d**) Comparison of Raman and CARS spectra of GNPs at 2400 to 3200 cm^−1^, 2D peak in Raman spectrum (2595 cm^−1^), and its shoulders disappear in the CARS spectrum, but two new peaks of 1150 cm^−1^ (considered to be a combination of D+D_1_ (approximately 1150 cm^−1^) modes) and 2960 cm^−1^ (considered to be 2GCARS) appeared in its CARS spectrum. Reproduced from [101]. (**e**) CARS spectral sweep of graphene at 1330 cm^−1^ Raman wavenumber. Reproduced from [95] (**f**) Image of h-BN under optical microscope, SRS, CARS, and SHG. (**g**) Raman and SRS scanning image and calibration spectrum of the same h-BN. Reproduced from [48], with permission from American Chemical Society, 2019. (**h**) Comparison of normalized CARS spectra of two-layer and three-layer graphene. (**i**) Classification result of test data by support vector machine (SVM) classifier, the percentage above the column is the accuracy rate.

**Figure 14 nanomaterials-10-02263-f014:**
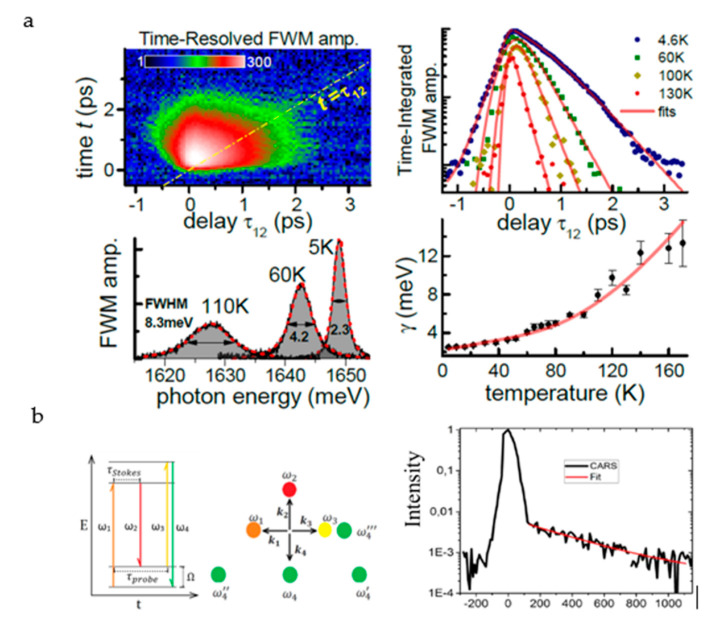
(**a**) The temperature-dependent exciton phase shift which was measured in the MoSe_2_ heterostructure region dominated by homogeneous broadening. Reproduced from [103], with permission from American Physical Society, 2020. (**b**) Schematic diagram of time-resolved CARS principle and decay graph of CARS signal with delay time. Reproduced from [108], with permission from American Chemical Society, 2017.

**Figure 15 nanomaterials-10-02263-f015:**
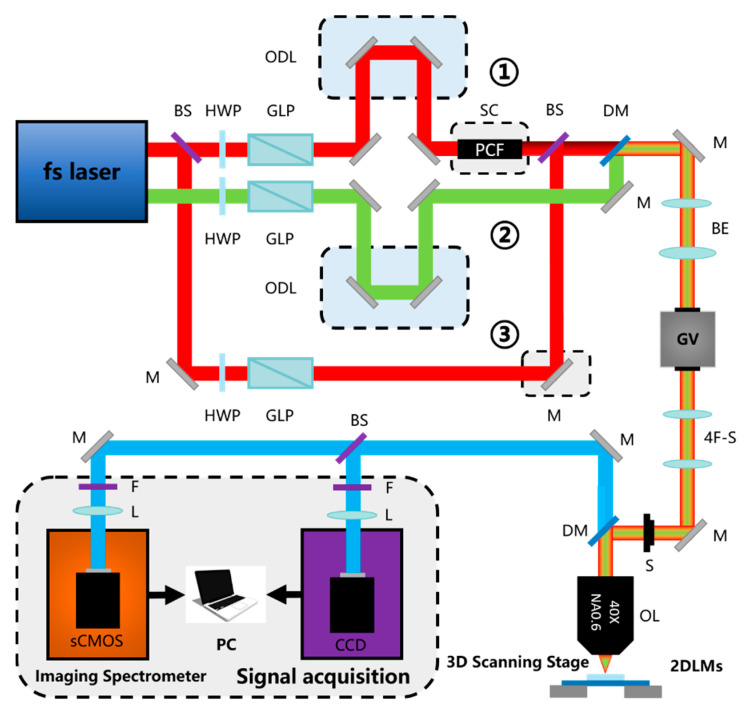
System instrument of our group. HWP: half-wave plate; GLP: Glan-Laser polarizer; ODL: optical delay line; PCF: photonic crystal fiber; SC: supercontinuum; DM: dichroic mirror; BS: beam splitter; GV: galvanometer; 4F-S: 4F system; sCMOS: scientific complementary Metal-Oxide-Semiconductor detector; BE: beam expander; OL: objective lens; F: filter.

**Table 1 nanomaterials-10-02263-t001:** Advantages and disadvantages of different Raman-based imaging techniques. Reproduced from [26], with permission from Taylor and Francis Group LLC (Books) US, 2020.

	Advantages	Disadvantages
Single-frequency CARS	High speed (us/pixel)	No spectral information
	Simple detection at new frequency	Nonresonant background
	Efficient backward detection	Background from fluorescence
Single-frequency SRS	High speed (us/pixel)	No spectral information
	No nonresonant background	Complicated detection
	Linear to molecular concentration	Accompanied by other pump-probe contrasts
	Phase matched	
	Not sensitive to incoherent background	
Multiplex CARS or SRS	Spectrally resolved detection	Integration time (tens of ms/pixle)
	Background removed in post-processing	
Hyperspectral CARS or SRS	Spectrally resolved detection	Narrow spectral window (200 cm^−1^)
	Fast acquisition (ms/pixel)	
Spontaneous Raman	Shot noise limited	Long integration time (s/pixle)
	Cost-effective (cw laser)	Very sensitive to incoherent background
	Whole spectrum analysis	

**Table 2 nanomaterials-10-02263-t002:** CARS bands of graphene nanoplatelets (GNPs), graphene oxide (GO), and highly oriented pyrolytic graphite (HOPG). Reproduced from [101]. (* means no test,/means there is no such characteristic peak).

Assignment	GNP (cm^−1^)	GO (cm^−1^)	HOPG (cm^−1^)
D	1300	1306	/
New band	/	1419	/
New band	1500	1516	/
G	1555	1584	1587
D’	/	/	*
2D(G’)	/	*	*
D + D_1_	2460	*	*
2G _CARS_	2960	*	*

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
