# Peer review of "Nonlinear Optical Characterization of 2D Materials"

_nanomaterials, 2020, doi:10.3390/nano10112263_

Round 1
Reviewer 1 Report
The authors are to be congratulated for a comprehensive review of a burgeoning field. This is a useful and timely piece of work and I recommend publication in Nanomaterials subject to minor revision.
Specifically:
The overall standard of English is generally good but there are enough incongruities that it can be irritating to read at times. I recommend that the authors make use of an English editing service. A few examples (of many possible):
Page 1 "Andre Geim" not "AndreGeim", "Scotch" not "scotch" (Scotch is a brand name of the product, not a generic name)
Line 187: "This process is an example of an optical parametric process" not "This process belongs to the optical parametric process"
Line 288: "a forward-detected CARS (F-CARS) microscope" not "forward-detected CARS (F-CARS) microscope"
Line 882: "4. Discussion" not "4. Discussions"
Many occurrences: "research" not "researches"
This list is by no means exhaustive.
The authors are in love with acronyms! As this review is more likely to be dipped into rather than read in its entirety, a list of acronyms should be provided. They should also consider whether they are needed: eg GO for graphene oxide is only used half-a-dozen times or so. (Note also that graphene is not capitalised unless it is the first word in a sentence).
These are all minor issues, overall this is an excellent review and will be useful to anyone interested in 2D materials.
Reviewer 2 Report
The authors provide a complete review of nonlinear optical analytical tools to deduce physical and chemical properties of 2D materials. Since these materials have benefited from intense attention the past decade thanks to their unique properties relevant for various applications, having sensitive tools and standardized analytical methods to our disposal is of great importance. In this sense, this review paper is fulfilling this requirement.
After introducing the different non-linear optical processes and derived techniques, authors introduce for each targeted physical or chemical property, the necessary state of the art and commonly used techniques with their limitations and what NLO can bring for a complete investigation of 2D materials. Authors give a very good review of existing NLO characterization techniques (principle and setup) with useful recommendations of the most efficient techniques depending on the targeted parameter. Authors also provide a complete overview of relevant physical and chemical properties of 2D materials that are commonly investigated to evaluate their performance for various applications. Authors even suggest a versatile NLO setup to perform multimodal analysis, necessary to improve accurary and perform quantitative measurements.
This review paper can be considered for publication in this special issue "Laser-Based Synthesis, Processing, and Characterization of 2D Quantum Materials" in Nanomaterials after adressing the minor suggestions listed below:
- The structure of the manuscript is very good and clear. However, a table of content is missing in the beginning to help the reader having an overview of the paper.
- Authors should highlight the fact that NLO becomes very convenient when coupled to microscopy to image 2D materials. The need for fast multiplex (space and energy) is not clearly stated in the abstract and introduction. As almost all NLO techniques are optical microscopy techniques, it would be better to clearly define it, perhaps in the title or at least in abstract, introduction and conclusion.
- Part 2 : The title is too general. It is fitting for the beginning of this part but after authors describe ONL microscopy techniques. Part 2 title could be: "Principle of NLO microscopy techniques”
- Part 2.2.1 Why not introducing SFG as a vibrational spectroscopy technique (already used for TMD). Nothing about SHG/SFG microscopy?
- Part 3.5: What about vibrational SFG for chemistry specificity?
- Part 4.1: this part is more summarizing the article than providing a proper discussion of the advantage of each technique for specific characterization.
- Part 4.5: An example from literature to highlight the importance of machine learning is missing here.
- Part 4.6: Why evaluate CRS future development regarding to SHG technique already widely used to characterize 2D materials. SHG is a very versatile for 2D mat: is there possible improvements of the technique?
- Discussion/conclusion: Does the NLO microscopy community targets to go towards sub-wavelength resolution, using local field enhancement techniques like in TERS? If yes, can the author address this point in the discussion or conclusion?
Minor points:
- Figure 1: Valleytronics is not introduced or discussed through out the paper.
- Figure 9: With what electronic state one of the laser is resonant in these SFG measurements?
- Figure 12 and 14: quality should be improved
- Line 128-131: ref?
- Line 161: Can be written differently: “is one of the kind of...”
- Line 198: Do author means TERS?
- Line 480: “As we all know” not necessary.
- Line 496: “2.2V => 2.2 V”
- Line 524: “the” instead of “our”
- Line 584, caption Fig 8: “Figure8” => “Figure 8” and “(c)Left” => “(c) Left”
- Line 665, caption Fig 10: “MoS2” = “MoS2”
- Line 672-673: in contradiction with previous sentence
- Line 685-686: “used for strain detection of 2D materials” is not necessary
- Line 747, Fig. 12 caption: “Figure12” => “Figure 12”
- Line 791, Fig. 13 caption: “Figure13” => “Figure 13”
- Line 795-798, Fig 13 caption: space are missing between wavenumber values and units
- Line 821: “selection rules” instead of “rules”
- Line 848: ref?
- Line 867, Fig. 14 caption: “Figure14” => “Figure 14”
- Line 882: Font of part 4 (discussions) is not correct
- Line 889, 907, 909, 914: part 3 instead of chapter 3
- Line 925-927: The sentence is not correct
- Line 927: What is super chemical specificity? Not appropriate term
- Line 929-930: vibrational SFG is also chemical specific and not discussed by authors.
- Line 954-958: too much “characterize”
- Line 998-1001: Sentence not clear
- Line 1056: “[119].” => “[119] .”
